# ZMYND10 functions in a chaperone relay during axonemal dynein assembly

Girish R Mali[1†‡], Patricia L Yeyati[1†], Seiya Mizuno[2], Daniel O Dodd[1], Peter A Tennant[1], Margaret A Keighren[1], Petra zur Lage[3], Amelia Shoemark[4], Amaya Garcia-Munoz[5], Atsuko Shimada[6], Hiroyuki Takeda[6], Frank Edlich[7,8], Satoru Takahashi[2,9], Alex von Kreigsheim[5,10], Andrew P Jarman[3], Pleasantine Mill[1]*

[1]MRC Human Genetics Unit, Institute of Genetics and Molecular Medicine, University of Edinburgh, Edinburgh, United Kingdom; [2]Laboratory Animal Resource Centre, University of Tsukuba, Tsukuba, Japan; [3]Centre for Discovery Brain Sciences, University of Edinburgh, Edinburgh, United Kingdom; [4]Division of Molecular and Clinical Medicine, University of Dundee, Dundee, United Kingdom; [5]Systems Biology Ireland, University College Dublin, Dublin, Ireland; [6]Department of Biological Sciences, University of Tokyo, Tokyo, Japan; [7]Institute for Biochemistry and Molecular Biology, University of Freiburg, Freiburg, Germany; [8]BIOSS, Centre for Biological Signaling Studies, University of Freiburg, Freiburg, Germany; [9]Department of Anatomy and Embryology, Faculty of Medicine, University of Tsukuba, Tsukuba, Japan; [10]Edinburgh Cancer Research UK Centre, Institute of Genetics and Molecular Medicine, University of Edinburgh, Edinburgh, United Kingdom

*For correspondence: Pleasantine.Mill@igmm.ed.ac.uk

†These authors contributed equally to this work

Present address: ‡MRC Laboratory of Molecular Biology, Cambridge, United Kingdom

Competing interests: The authors declare that no competing interests exist.

**Abstract** Molecular chaperones promote the folding and macromolecular assembly of a diverse set of 'client' proteins. How ubiquitous chaperone machineries direct their activities towards specific sets of substrates is unclear. Through the use of mouse genetics, imaging and quantitative proteomics we uncover that ZMYND10 is a novel co-chaperone that confers specificity for the FKBP8-HSP90 chaperone complex towards axonemal dynein clients required for cilia motility. Loss of ZMYND10 perturbs the chaperoning of axonemal dynein heavy chains, triggering broader degradation of dynein motor subunits. We show that pharmacological inhibition of FKBP8 phenocopies dynein motor instability associated with the loss of ZMYND10 in airway cells and that human disease-causing variants of ZMYND10 disrupt its ability to act as an FKBP8-HSP90 co-chaperone. Our study indicates that primary ciliary dyskinesia (PCD), caused by mutations in dynein assembly factors disrupting cytoplasmic pre-assembly of axonemal dynein motors, should be considered a cell-type specific protein-misfolding disease.
DOI: https://doi.org/10.7554/eLife.34389.001

## Introduction

Macromolecular motors of the dynein family power the essential beating of motile cilia/flagella. Motile cilia propel sperm cells, generate mucociliary clearance in airways, modulate nodal flow for embryonic left-right patterning and circulate cerebrospinal fluid inside the brain. Force-generating dynein motors are large molecular complexes visible by transmission electron microscopy (TEM), as 'outer' and 'inner dynein arms' (ODA, IDA) spaced at regular intervals along the microtubule axoneme. Each ODA motor consists of catalytic heavy chains (HC), intermediate chains (IC) and light chains (LC); IDAs have a more heterogeneous composition. In mammals, at least 4 ODA and 7 IDA

subtypes exist, each containing different HCs (*Kollmar, 2016*; *Wickstead and Gull, 2007*). Defective dyneins render cilia immotile, resulting in the severe congenital ciliopathy in humans termed Primary Ciliary Dyskinesia (PCD, OMIM: 242650). Understanding the molecular causes of PCD requires addressing how complex molecular machines like the dyneins get built during cilium biogenesis.

PCD-causing mutations are most frequently detected in genes encoding structural ODA subunits such as the intermediate chains (*DNAI1* and *DNAI2*; [*Pennarun et al., 1999*; *Guichard et al., 2001*; *Loges et al., 2008*]), or the catalytic heavy chain (*DNAH5*; [*Olbrich et al., 2002*]), all of which disrupt motor assembly and/or function. Consequently, mutant multiciliated cells form cilia but these fail to move, lacking ODAs by TEM or immunofluorescence.

In addition to these structural subunits of dynein motors, several PCD-causing mutations are also found in a newly discovered set of genes, the 'dynein axonemal assembly factors' (DNAAFs) whose functions are poorly understood. DNAAFs are proposed to assist heat shock protein (HSP) chaperones to promote subunit folding and cytoplasmic pre-assembly of dynein motors. DNAAFs are presumed to act as cilial-specific co-chaperones based on proteomic identification of interactions with both 'client' dynein chains and canonical chaperones. Of the known assembly factors, KTU/DNAAF2 and DYX1C1/DNAAF4 have the most direct biochemical links to HSP90 and HSP70 chaperones, as well as ODA intermediate chain DNAI2 (IC2) (*Omran et al., 2008*; *Tarkar et al., 2013*). Additionally, KTU/DNAAF2, PIH1D3/DNAAF6 and SPAG1 share structural domains with key non-catalytic subunits of the R2TP-HSP90 chaperoning complex, PIH1 and TAH1, respectively. R2TP is a well established HSP90 co-chaperone which confers specificity during the assembly of multisubunit enzymes (*Pal et al., 2014*). DYX1C1/DNAAF4 and LRRC6 each have a CS or p23-like domain (*Kott et al., 2012*; *Tarkar et al., 2013*), p23 being a well characterised HSP90 co-chaperone acting during the last steps of the HSP90 chaperone cycle (*Li and Buchner, 2013*). Interactions between LRRC6, DNAAF1/LRRC50 and C21ORF59/Kurly/CFAP298 were also recently reported which, coupled with the phenotypic analysis of *Lrrc6* mutant mice, suggests that these assembly factors may function together in a late-acting complex (*Inaba et al., 2016*; *Jaffe et al., 2016*). The functions of DNAAF3 and DNAAF5/HEATR2 which have no reported links to chaperones, remain elusive (*Diggle et al., 2014*; *Mitchison et al., 2012*). Altogether, the current view is that many DNAAFs transiently participate as HSP90 co-chaperones during the macromolecular assembly of dynein motors before they are finally transported into the cilia.

Dynein pre-assembly has been well studied in unicellular eukaryotes such as *Chlamydomonas* and *Paramecium*. For ODAs in *Chlamydomonas*, affinity purification confirmed all three HCs (HCs; α, β, and γ, each of ~500 kDa) and two ICs (IC1, 78 kDa; IC2, 69 kDa) are pre-assembled as a three headed complex and exist in a cytoplasmic pool prior to ciliary entry (*Fok et al., 1994*; *Fowkes and Mitchell, 1998*; *Qin et al., 2004*). This cytoplasmic pre-assembly pathway is highly conserved and exists in all ciliated eukaryotes (*Kobayashi and Takeda, 2012*). While it is clear that the aforementioned assembly factors aid axonemal dynein pre-assembly, their precise molecular functions within the pre-assembly pathway still remain largely unknown.

Previous studies had established a strong genetic link between loss of ZMYND10, a putative DNAAF, and perturbations in dynein pre-assembly (*Cho et al., 2018*; *Kobayashi et al., 2017*; *Kurkowiak et al., 2016*; *Moore et al., 2013*; *Zariwala et al., 2013*), however the molecular role of ZMYND10 as a DNAAF in this process remained unclear. In order to probe the mammalian dynein pre-assembly pathway in greater molecular and cellular detail, we generated *Zmynd10* null mice by CRISPR genome editing. We used different motile ciliated lineages at distinct stages of differentiation from our mammalian mutant model to ascribe a molecular role to ZMYND10 within the dynein pre-assembly pathway.

Our studies implicate a novel chaperone complex comprising of ZMYND10, FKBP8 and HSP90 in the maturation of dynein HC clients and provide novel evidence of the temporally restricted nature of interactions within this chaperone-relay system, later involving LRRC6 likely to promote stable inter-subunit interactions. We postulate that a chaperone-relay system comprising of several discrete chaperone complexes handles the folding and stability of distinct dynein subunits all the while preventing spurious interactions during cytoplasmic pre-assembly.

## Results

### Generation of a mammalian PCD model to characterize dynein assembly

We targeted exon 6 of mouse *Zmynd10* to target all predicted protein isoforms, with three guide RNA (gRNA) sequences for CRISPR genome editing and generated several founders with insertion, deletion and inversion mutations (*Figure 1A*, *Figure 1—figure supplement 1*). Null mutations from the different CRISPR guide RNAs gave identical phenotypes, confirming the phenotypes are due to loss of ZMYND10, as opposed to off-target effects. For detailed analysis, we focused on a −7 bp deletion mutant line (*Zmynd10* c.695_701 p.Met178Ilefs*183), which results in a frame shift with premature termination. Generation of a null allele was verified by ZMYND10 immunoblotting of testes extracts (postnatal day 26, P26) and immunofluorescence of multiciliated ependymal cells and lung cryosections (*Figure 1B–D*)

*Zmynd10* mutant mice displayed several clinical features of PCD including heterotaxia, progressive hydrocephaly and chronic mucopurulent plugs in the upper airways, all features consistent with defects in ciliary motility (*Figure 1E–H*, *Figure 1—figure supplement 2*). This was directly confirmed by high-speed video microscopy of ependymal cells, where cilia of normal number and length were present but failed to move (*Videos 1*, *2*, *3* and *4*, *Figure 1—figure supplement 3*). Ultrastructure analysis of tracheal cilia axonemes revealed an absence of both outer and inner dynein arms (*Figure 1F*). The hydrocephaly phenotype was particularly pronounced on a C57BL6/J background and the majority of mutants died around weaning (P17-P21). On outbred backgrounds, male infertility and sperm immotility were also noted in homozygous mutant animals (*Figure 1—figure supplement 2*, *Videos 5* and *6*). These findings demonstrate that ZMYND10 functions are exclusively required in most motile ciliated cell lineages.

### Mis-assembled dynein motors are blocked from entering cilia and cleared in *Zmynd10* mutants

We analysed expression of dynein subunits in different postnatal tissues by immunofluorescence and immunoblotting from *Zmynd10* mutants, focusing on ODA components for which the most robustly validated immunoreagents exist, to assess whether ZMYND10 loss impacts ODA levels. In adult trachea (P26) and oviducts (P26-30), total levels of the ODA HCs DNAH9 and DNAH5, as well as ICs DNAI1 and DNAI2 were reduced by immunoblot (*Figure 2A,B*) and immunofluorescence (*Figure 2C,D*). In agreement with recent studies(*Cho et al., 2018*), no alteration in dynein transcripts were detected by RT-qPCRs on mutant oviduct (P12) total RNA, supporting the findings that the zinc-finger MYND domain of ZMYND10 plays cytoplasmic molecular scaffold functions other than a nuclear transcriptional role (*Figure 2E*). Critically, immunoblots of P7 *Zmynd10* mutant oviduct lysates, an early stage corresponding to synchronized multicilial axonemal elongation (*Dirksen, 1974*), showed a laddering of DNAH5 products using an antibody raised against an N-terminal epitope, indicating post-translational destabilization of DNAH5 in the absence of ZMYND10 (*Figure 2F*).

These observations raise the possibility that in the absence of ZMYND10, individual dynein subunits are initially synthesised during cytoplasmic pre-assembly before a quality-control response is triggered to clear stalled dynein motor assembly intermediates that fail to reach the mutant axonemes. The process of mammalian cytoplasmic pre-assembly in terms of dynamics of localization and levels of dynein subunits has been previously documented (*Diggle et al., 2014*). 'Immature' cells stain strongly for dynein subunits within the cytoplasm, they appear rounder and their cilia are apparently shorter than those cells in which dynein subunits exclusively stain the ciliary compartment. Using Sentan as an independent marker for 'mature' motile cilia, we were able to detect a strong, focused signal at the tips of cilia with exclusively ciliary staining of axonemal dyneins (*Figure 3A*, lower left inset). Sentan is a component of the apical ciliary crown structure of mature motile cilia, where peripheral singlet microtubules are capped by electron dense material abutting the membrane (*Kubo et al., 2008*). In contrast, in cells without clear apical Sentan signal, shorter cilia and cytoplasmic staining (DNAI2, DNALI1) were observed, consistent with these cells being 'immature' and in the process of assembling dynein motors and cilia (*Figure 3A*, upper right inset). We therefore postulate that these dynamic patterns of dynein subunits represent 'mature' and 'immature'

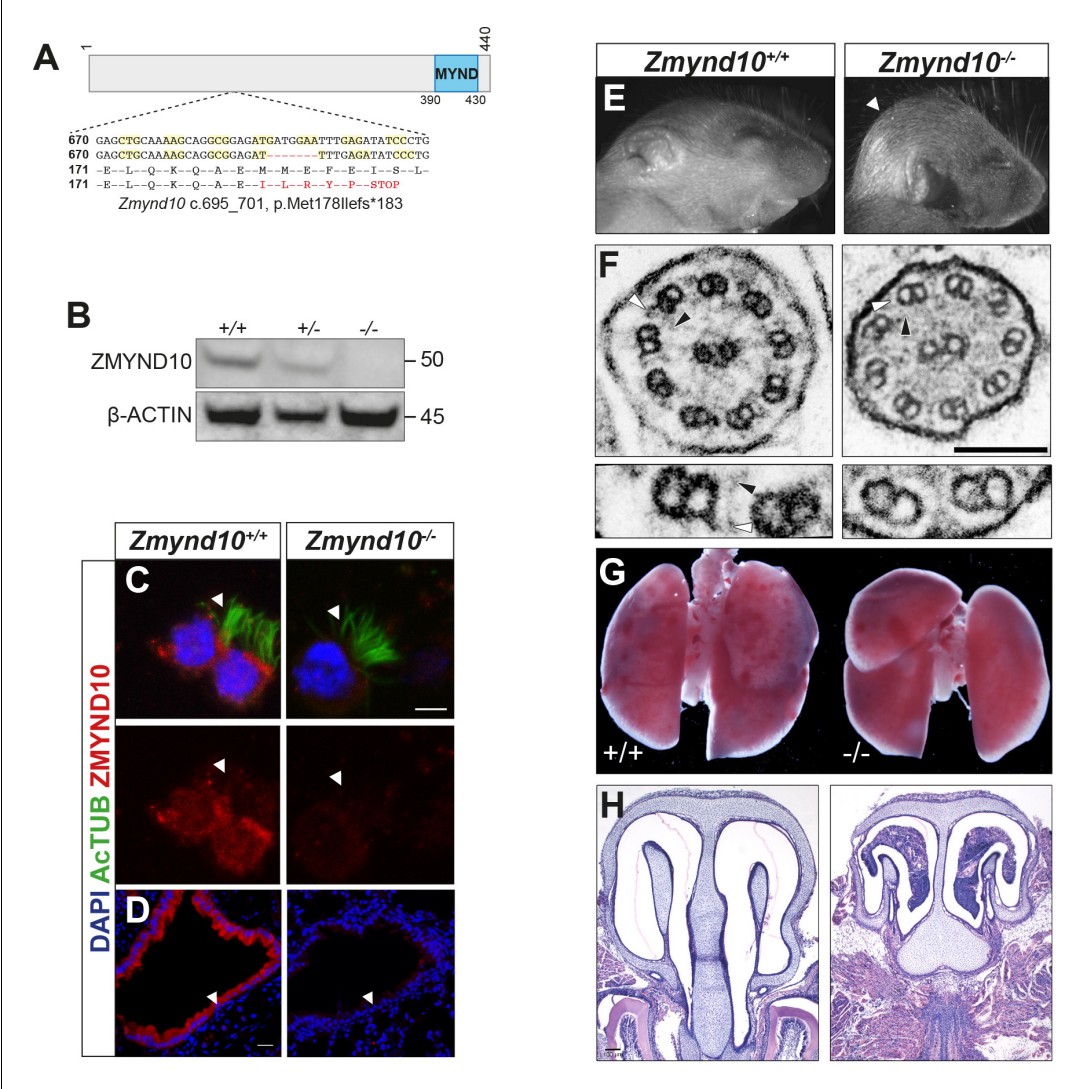

**Figure 1.** Loss of *Zmynd10* in mice results in a PCD phenotype. (**A**) Schematic illustrating the null allele generated by a −7 bp CRISPR deletion in *Zmynd10* exon 6. (**B**) Immunoblots from testes extracts from postnatal day 26 (P26) control and mutant male mice show loss of ZMYND10. (**C, D**) Immunostaining for ZMYND10 reveals a complete loss of signal in multiciliated ependymal cells (**C**) and lung cryo-sections (**D**). Multicilia are marked with acetylated α- tubulin (**C**). (**E**) Neonatal *Zmynd10* mutants display hydrocephaly; the white arrowhead points to doming of the head. See also *Figure 1—figure supplement 2E*. (**F**) TEM of tracheal ciliary axoneme cross-sections shows a lack of axonemal outer (ODA: white arrowhead) and inner (IDA: black arrowhead) dynein arms in mutants. (**G**) Representative image of a gross dissection of lungs shows *situs inversus totalis* in mutants. See also *Figure 1—figure supplement 1F*. (**H**) H&E staining of coronal sections of nasal turbinates reveals mucopurulent plugs in mutants. Scale bars in (**C**) = 5 μm, in (**D**) = 100 μm, in (**F**) = 100 nm.

DOI: https://doi.org/10.7554/eLife.34389.002

The following figure supplements are available for figure 1:

**Figure supplement 1.** Generation of *Zmynd10* mutant mice by CRISPR genome editing.
DOI: https://doi.org/10.7554/eLife.34389.003

**Figure supplement 2.** Detailed phenotypic analysis of *Zmynd10* mutants.
DOI: https://doi.org/10.7554/eLife.34389.004

**Figure supplement 3.** No gross ciliary defects are seen in *Zmynd10* mutants.
DOI: https://doi.org/10.7554/eLife.34389.005

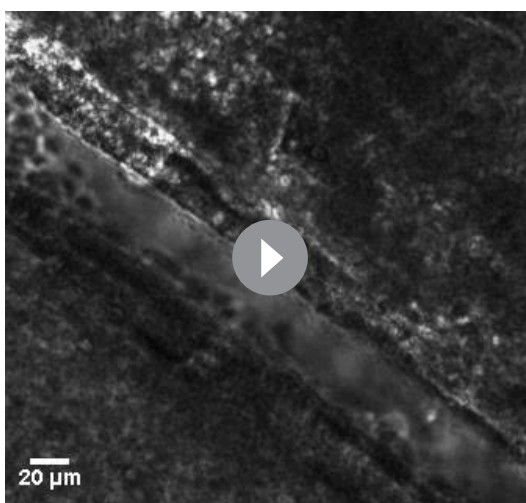

**Video 1.** Rapid ependymal ciliary motility in lateral ventricles of a wild type mouse. High-speed video microscopy on a coronal brain vibratome section (postnatal day 11 mouse, littermate control) shows ependymal cilia lining lateral ventricles beating with high frequency in a wild type mouse.

DOI: https://doi.org/10.7554/eLife.34389.006

cells during assembly of motile cilia, which we can see in cells isolated from nasal turbinates of control mice (3B middle and upper panels, respectively). In *Zmynd10* mutants, no apparent defects in ciliary length or number were observed (*Figure 1—figure supplement 3*) however outer or inner arm dyneins fail to incorporate into mature ciliary axonemes. Importantly, no cytoplasmic accumulations were noted in 'mature' ciliated cells (*Figure 3B* panels with arrows). Surprisingly, cytoplasmic staining was observed in both 'immature' control and *Zmynd10* mutant cells (*Figure 3B*, arrowheads). A similar staining pattern was observed for DNALI1 (*Figure 3C*) indicating ODA and IDA dynein subunit precursors are initially synthesised normally, further supporting that ZMYND10 loss does not impact their transcription or translation. Instead, loss of ZMYND10 leads to dyneins being robustly cleared when their pre-assembly is perturbed.

## ODA and IDA complexes are defective and unstable in the absence of ZMYND10

As cytoplasmic staining for DNAI2 and DNAH5 was detected in *Zmynd10*$^{-/-}$ immature respiratory cells, suggesting that they were initially synthesized, we sought to verify if they were assembled into complexes using the *in situ* proximity ligation assay (PLA). In control immature human nasal brush epithelial cells, we detected PLA signals consistent with DNAI2 and DNAH5 existing in both cytoplasmic and axonemal complexes (*Figure 4A*). However, we detected a highly reduced number of PLA positive foci in nasal epithelial cells of P7 *Zmynd10*$^{-/-}$ mice, with complexes restricted entirely to the cytoplasm in contrast to the strong axonemal staining observed in similarly staged controls (*Figure 4B*, *Figure 4—figure supplement 1*). To directly examine the interactions between ODA IC and HC subunits, we immunoprecipitated endogenous DNAI2 (IC2) from postnatal testes (P26), trachea (P7 and P90) and oviduct (P7) extracts from *Zmynd10*$^{-/-}$ animals. DNAI2 co-precipitated DNAI1 (IC1) at similar levels from both wild type and mutant P26 testes extracts (*Figure 4C*). This indicated that loss of ZMYND10 does not primarily impact IC subunit

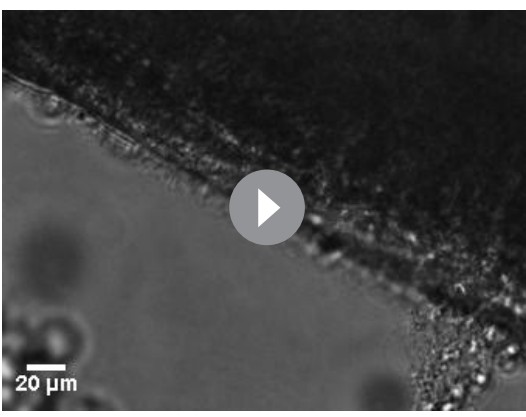

**Video 2.** Immotile ependymal cilia lining lateral ventricles of *Zmynd10* mutant mouse. High-speed video microscopy on a coronal vibratome section of a brain from a *Zmynd10* mutant mouse (postnatal day 11) shows complete loss of ependymal cilia motility.

DOI: https://doi.org/10.7554/eLife.34389.007

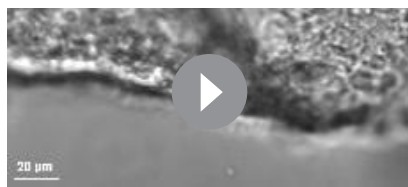

**Video 3.** Control murine ependymal cilia with metachronal waveform. High-speed video microscopy on a coronal vibratome section of a *Zmynd10* mild hypomorphic mutant mouse brain (p. M179del; postnatal day 24) shows arrays of cilia beating in a metachronal waveform and actively generating fluid flow to move particulates over the ventricle tissue.

DOI: https://doi.org/10.7554/eLife.34389.008

heterodimerization or stability during the assembly process. Similarly, the relative enrichment of DNAI2 changed very little between mutant and wild type P7 oviduct and trachea mutant extracts. In contrast, we observed significantly reduced co-immunoprecipitation of DNAH5 by DNAI2 in P7 oviduct and trachea mutant extracts (0.47 and 0.56 fold reduction respectively, normalized to total levels, *Figure 4D,E* and *Figure 4—figure supplement 1B,C*). Moreover, we observed similar degradative bands (arrowheads) for DNAH5 in the mutant samples indicating that any DNAH5 that is incorporated may be poorly folded and unstable, in the absence of ZMYND10. We hypothesize that this reduced association between the two subunits is due to the HC subunit being in an assembly incompetent, unstable state such that any substandard complex would be targeted for subsequent degradation (*Figure 4F*).

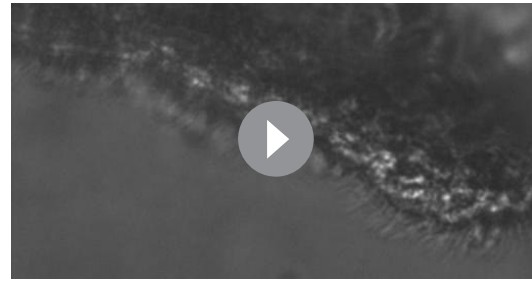

**Video 4.** Tufts of immotile ependymal cilia in *Zmynd10* mutant murine brain. High-speed video microscopy on a coronal vibratome section of a *Zmynd10* null mutant mouse brain (p. L188del; postnatal day 29) shows arrays of immotile cilia lining the ventricle tissue with no active fluid flow noticeable.
DOI: https://doi.org/10.7554/eLife.34389.009

ZMYND10 loss also leads to absent IDA motors from human, fly and mouse cilia (*Cho et al., 2018*; *Kobayashi et al., 2017*; *Kurkowiak et al., 2016*; *Moore et al., 2013*; *Zariwala et al., 2013*). To bypass the limitation of robust immunoreagents for IDA detection, we used label-free quantitative proteomics comparing postnatal testes extracts from P25 control and *Zmynd10* mutant littermates. ZMYND10 is highly expressed in the cytoplasm of round and elongating spermatids, as well as maturing sperm (*Figure 5A*). In the absence of ZMYND10, mature sperm form but lack expression of outer and inner dynein subunits (*Figure 5B*). At P25, we hypothesized that synchronized spermiogenesis and flagellar extension at this stage would correspond with cytoplasmic pre-assembly of flagellar precursors. Whilst protein expression profiles were not different between mutant and controls for differentiation, meiosis and cell death markers (*Supplementary file 1*), the expression profile for the motility machinery showed specific and significant changes wherein almost all the axonemal dynein HCs (outer and inner) detected were reduced whilst the other axonemal dynein subunits were generally not significantly changed (ICs WDR78 and DNAI1, DNAI2) (*Figure 5D*, *Figure 5—figure supplement 1*). This is distinct from previous observations in *Chlamydomonas*, where loss of DNAAFs (DNAAF1, 2 and 3) impacting HC stability generally led to an aberrant cytosolic accumulation of IC subunits (*Mitchison et al., 2012*), highlighting a key difference between the two model

systems. Components of the radial spokes (RS) and dynein regulatory complex (DRC) were also unchanged (*Figure 5D*). Interestingly, several DNAAFs including the co-chaperones DNAAF4 and DNAAF6 were moderately but significantly up regulated in *Zmynd10* mutants suggestive of a proteostatic response to counter aberrant pre-assembly as it progresses.

## A ZMYND10-FKBP8-HSP90 complex mediates maturation of dynein heavy chains

To further understand how ZMYND10 regulates stability of axonemal dynein HC subunits, we aimed to generate an endogenous ZMYND10 interactome in P30 mouse testes using two validated commercial ZMYND10 polyclonal antibodies followed by mass-spectrometry (AP-MS, *Supplementary file 2*). Overlapping interactors in the endogenous affinity purifications of

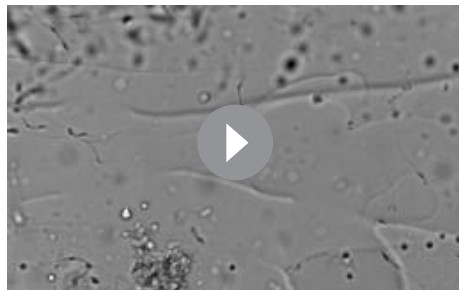

**Video 5.** Aberrant flagellar motility in *Zmynd10* mutant murine epididymal spermatozoa. High-speed video microscopy on mature spermatozoa extracted from the epididymis of a 5 month old *Zmynd10* CRISPR founder mutant mouse and slowed-down in methylcellulose. The majority of spermatozoa were completely immotile but rarely displayed highly aberrant flagellar movements as observed in the video.
DOI: https://doi.org/10.7554/eLife.34389.010

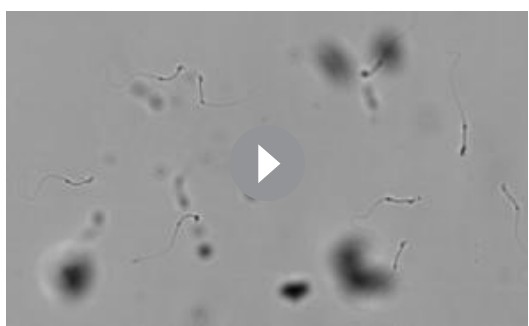

**Video 6.** Sinusoidal flagellar motility in wild type murine epididymal spermatozoa. High-speed video microscopy on mature epididymal spermatozoa extracted from a 5 month-old wild type mouse (littermate control) and slowed-down in methylcellulose. Virtually all spermatozoa underwent forward motion with the flagella displaying a sinusoidal beat pattern.

DOI: https://doi.org/10.7554/eLife.34389.011

ZMYND10 included ODA HCβ DNAH17 (fold enrichment: 5.8x (P), 2.9x (S); p>0.05), HSP90 (1.5x, 1.8x; p>0.01) and the well-characterized immunophilin FKBP8 (FK506-binding protein (FKBP) family member) (6.2x, 8.2x; p>0.001), which can act as an HSP90 co-chaperone (*Figure 6A*).

This suggested a novel association between a mammalian DNAAF, putative 'client' dynein heavy chain and the ubiquitous HSP90-FKBP8 chaperone complex during cytoplasmic pre-assembly in vivo. To validate these interactions, we immunoprecipitated endogenous ZMYND10 and FKBP8 from control versus mutant testes samples (P30: *Figure 6B*) and cultures of differ-entiating human tracheal epithelial cells (D17 ALI1: *Figure 6C*), both during cytoplasmic pre-assembly. Indeed, we confirmed that ZMYND10 interacted with FKBP8 and HSP90, but not with other DNAAFs, including previously identified interactor LRRC6 (*Figure 6B*). We also corrobo-rated associations between FKBP8, ODA HCα DNAH5, HSP90 and ZMYND10 using human Air-Liq-uid Interface (ALI) cultures and P7 oviducts (*Figure 6C,D*), as we failed to find specific immunoreagents against testes ODA HC isoforms (i.e. ODA HCβ DNAH17 or HCα DNAH8). Taking step-wise ODA macromolecular assembly into account, we found that whilst DNAI1 co-immunopre-cipitated DNAH5, it did not immunoprecipitate either ZMYND10 or FKBP8 (human ALI tracheal cells D17: *Figure 6C*). Together our data suggests that the DNAI1-DNAH5 interaction occurs in a com-plex that is distinct and downstream from the FKBP8-DNAH5-ZMYND10 complex, as supported by our DM-CHX experiments in which both subunits are destabilized after 24 hr drug treatment (see below).

We verified the interactions between ZMYND10 and FKBP8 as well as HSP90 using tagged ZMYND10 affinity purifications from primary ciliated HEK293 cells (*Figure 7B*). In agreement, direct interaction between the N-terminus of FKBP8 and MYND domains of ZMYND6/PHD2 (*Barth et al., 2009*) and ZMYND20/ANKMY2 (*Nakagawa et al., 2007*) have also been reported. To characterize the complex between FKBP8 and ZMYND10 further, we mapped their interaction interface and assessed its potential biological relevance by generating point mutations in ZMYND10 (*Figure 7A*). The W423A mutation falls within the MYND domain and is predicted to functionally disrupt one of two $Zn^{2+}$-fingers in the MYND domain. Located just before the MYND domain, the T379C PCD patient mutation (*Zariwala et al., 2013*) failed to disrupt binding to LRRC6, suggesting some other underlying pathogenic mechanism exists for this mutation, one which we hypothesized could involve FKBP8. Affinity purification of ZMYND10-turboGFP variants from HEK293 cells revealed that both point mutations abolished endogenous FKBP8 binding (*Figure 7B*). These results indicate that the interaction interface for FKBP8 extends beyond the MYND domain of ZMYND10, consistent with recent deletion mutations described in *medaka* capable of functional rescue (*Kobayashi et al., 2017*) and suggest that loss of ZMYND10-FKBP8 interaction may underlie the pathogenic effect of the T379C PCD mutation.

FKBP8 is a peptidyl-prolyl isomerase (PPIase), which catalyzes cis-trans isomerization of proline peptide groups and is one of the rate-determining steps in protein folding. To test whether its PPIase activity is critical for stabilization of dynein HCs, we treated immature day 17 and mature day 60 (D17 and D60 ALI) human tracheal epithelial cell cultures with a specific PPIase inhibitor DM-CHX (*Edlich et al., 2006*) for 24 hr and assayed extracts for stability of ODA subunits by immunoblotting. Immature cultures were very sensitive to FKBP8 inhibition, where cytoplasmic levels of DNAH5 were reduced to ~10% of control levels after DM-CHX (150 µM). In mature cells, fully assembled com-plexes within cilial axonemes were less sensitive to DM-CHX treatment (*Figure 7C and D*). Surpris-ingly, a very striking destabilization of DNAI1 was also observed in immature cultures under these

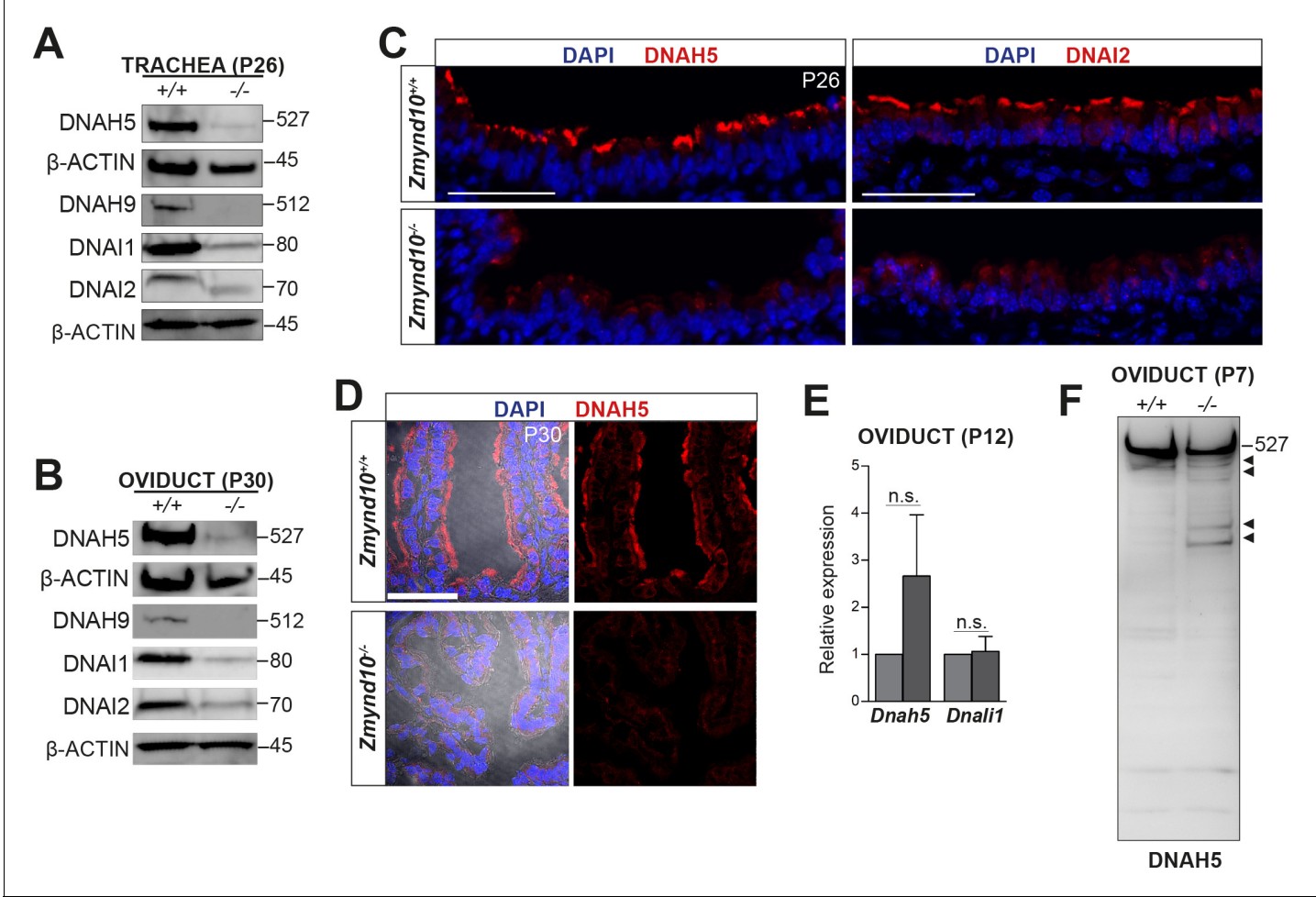

**Figure 2.** Global post-transcriptional destabilization of dyneins occurs in *Zmynd10* mutant motile ciliated tissues. Immunoblots of whole protein extracts from trachea P26 (**A**) and P30 oviducts (**B**) show reduced abundance of ODA subunits HC-γ, HC-β (DNAH5 and DNAH9), IC1 and IC2 (DNAI1 and DNAI2). Immunofluorescence of trachea (**C**) and oviduct (**D**) tissue sections show loss of axonemal DNAH5 and DNAI2 staining as well as reduced total abundance in *Zmynd10* mutants compared to controls. Brightfield is included in oviduct merge panel (**D**) to highlight absence of staining in cilia in *Zmynd10* mutants. Scale bars in (**C**) and (**D**) = 50 μm. (**E**) No significant changes are detected in levels of dynein transcripts by quantitative RT-PCR of (*Dnah5, Dnali1*) normalized to (*Tbp*) in P12 *Zmynd10* mutant oviducts (n = 3/genotype, dark grey *Zmynd10* mutants). (**F**) During early motile ciliogenesis, mildly reduced levels and laddering consistent with degradative, misfolded intermediates (arrowheads) of ODA HC-γ DNAH5 are detected in *Zmynd10* mutant oviducts (P7). These will be subsequently cleared as tissue differentiation proceeds.
DOI: https://doi.org/10.7554/eLife.34389.012

conditions. This supports the possibility that a transient requirement of the PPIase activity of FKBP8 is necessary for the folding and/or stability of axonemal dyneins in the cytoplasm.

Immunofluorescence of mouse tracheal epithelial cells (mTECs) shows that treating parallel cultures once they have differentiated (D3 ALI, 14 days vehicle followed by 7 days DM-CHX), or as they begin to differentiate (D3 ALI, 14 days DM-CHX followed by 7 days vehicle) have drastically different outcomes (*Figure 7E*). These results are in agreement with immunoblot data supporting the findings that inhibiting the PPIase activity of FKBP8 in mature ALI cultures does not affect motility or levels of DNAH5 or DNAI2, whilst early inhibition can drastically reduce the levels of these ODA components abolishing motility (*Figure 7E*).

## A successive set of complexes are involved in multiple steps of axonemal dynein assembly

In the chaperone cycle of HSP90, PPIs like FKBP8 are part of the intermediate stage where HSP90 is bound to the client protein and ATP. Co-chaperones containing a p23-like domain enter at the last

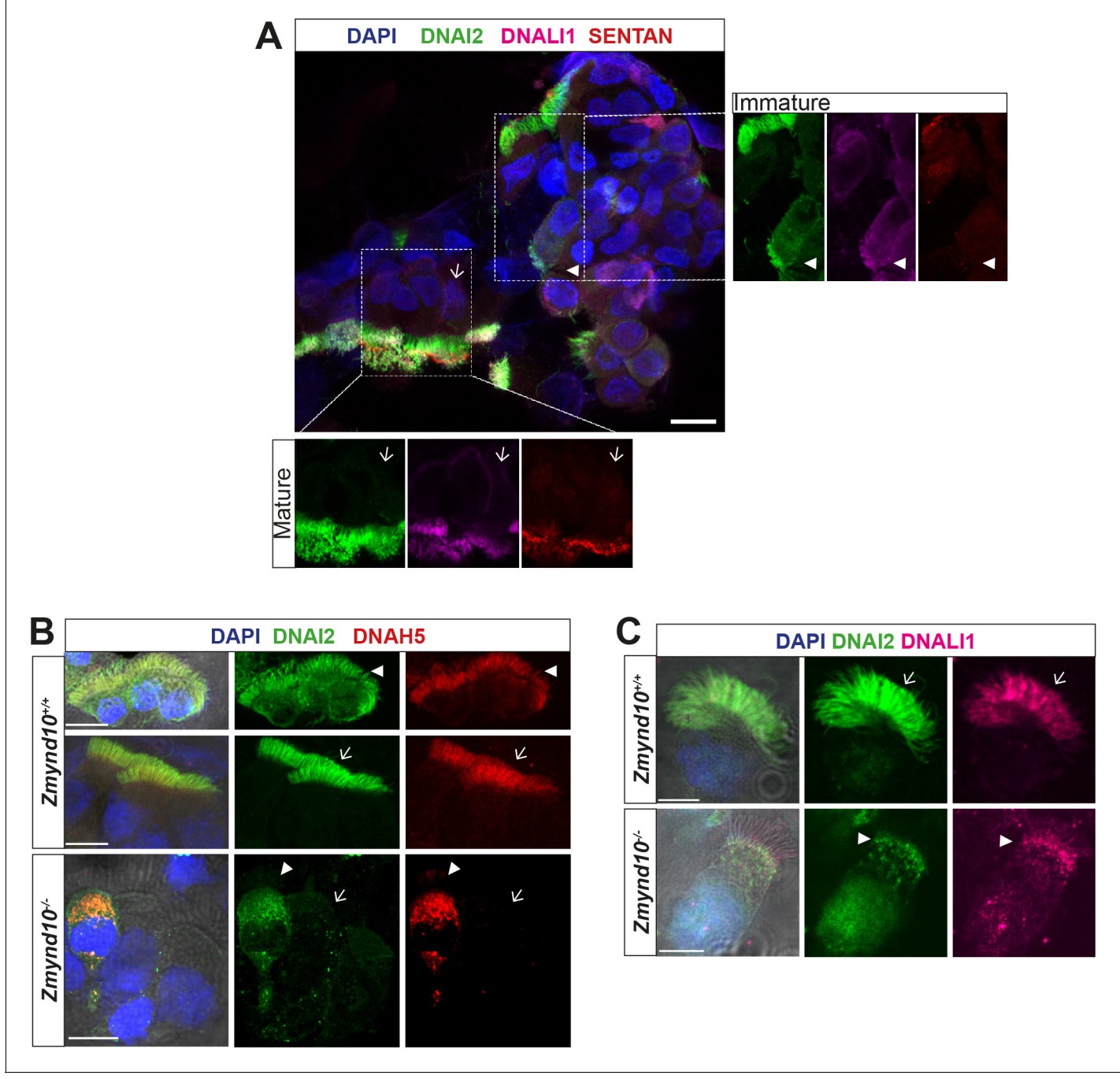

**Figure 3.** Loss of ZMYND10 perturbs sub-cellular distribution and levels of dynein complexes during pre-assembly. (A) Z-projection through healthy human donor nasal brush immunofluorescence shows a mix of 'mature' motile ciliated cells (lower inset, arrow) with exclusively axonemal staining of dynein subunits (DNAI2 green, DNALI1 magenta) and strong foci of SENTAN (red) at cilial tips, as well as 'immature' cells having cytoplasmic staining of dynein subunits and no SENTAN (right inset, arrowheads). Scale bar = 10 μm. (B) Nasal brush immunofluorescence from *Zmynd10* mice shows components of outer arm dyneins (DNAH5, DNAI2) are initially expressed in apical cytoplasm of immature mutant cells (arrowheads) but subsequently undergo 'clearance' in mature cells (lower panels, arrows), whilst all complexes exclusively translocate into cilia in control mature cells (middle panels, arrows). (C) Inner arm dynein component DNALI1 is expressed and apically arrested in immature mutant cells (arrowhead) and unlike controls, will get subsequentlycleared along with the ODA IC, DNAI2, in mature mutant cells. Scale bars = 5 μm.

DOI: https://doi.org/10.7554/eLife.34389.013

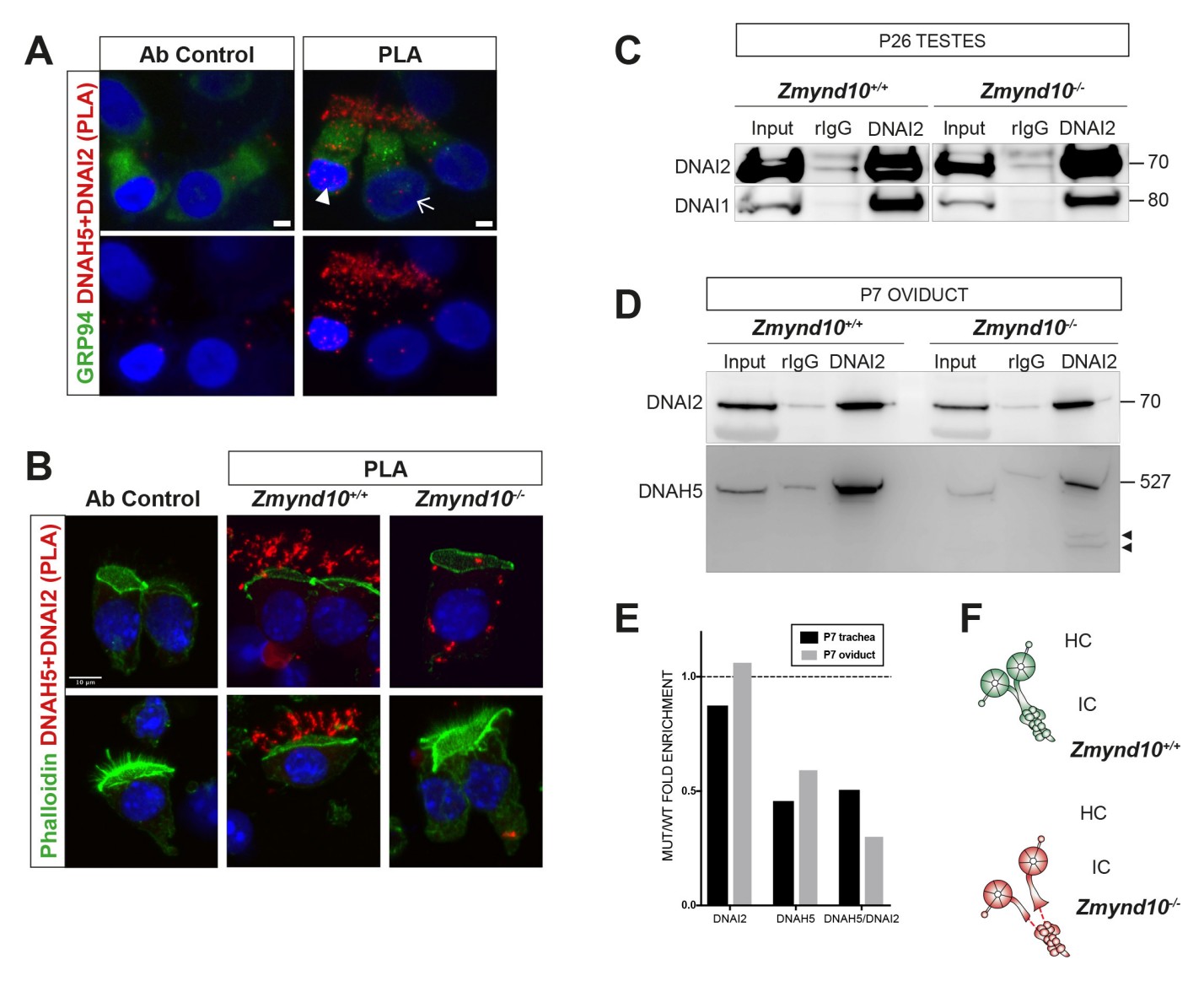

**Figure 4.** Sequential cytosolic assembly of outer arm dynein components occurs in mammalian motile ciliogenesis in a process requiring ZMYND10. (A,B) Z-projections of Proximity Ligation Assay (PLA) on human donor (A) or mouse P7 (B) nasal brush biopsies confirming ODA subunits (mouse IC2/DNAI2 and HC-γ/DNAH5) are pre-assembled in the cytoplasm of mammalian multiciliated cells. Control (antibody only control, Ab control) sections incubated with only DNAH5 show no PLA signal. Red spots denote individual ODA complexes (<40 nm) that appear as peri-nuclear foci in immature cells (arrowhead) and translocate to cilia in mature cells (arrow). (B) In *Zmynd10* mutant cells, reduced number of foci are observed and restricted only to the cytoplasm, highlighting defects in cytoplasmic pre-assembly. GRP94 was used as a pan-cytosolic marker (A) or phalloidin for apical actin ring (B). Nuclei are stained with DAPI (blue). See also *Figure 4—figure supplement 1A* for mouse IC2/DNAI2 and HC-β/DNAH9 PLA. Scale bars in (A) = 5 μm and (B) = 10 μm. (C) Endogenous Immunoprecipitation of DNAI2 from P26 Testes extracts reveals no defects in DNAI2 association with its heterodimeric partner DNAI1 in *Zmynd10* mutants. (D) Endogenous Immunoprecipitation of DNAI2 from P7 oviduct extracts show disruption in subsequent association between DNAI2 and DNAH5 in mutants compared to controls as quantified by intensities of the DNAH5 pull-down bands (E). Arrowheads show predicted degradative or misfolded intermediates of DNAH5 polypeptide in the mutants only. Numbers to the right of panels denote protein molecular weight in kDa. See also *Figure 4—figure supplement 1B,C* for analysis in P7 and P90 trachea. (E) Ratio of mutant versus wild type fold enrichment (from 4D, *Figure 4—figure supplement 1B*: IP/input) for DNAI2 and DNAH5, normalized for differences in stability in input, as well as amount of DNAH5/DNAI2 complexes. (F) Schematic of axonemal ODA showing the intermediate chain heterodimers (IC) bind normally to heavy chains (HC) to form the entire motor complex in controls (green) and that this association is perturbed in mutants (red).

DOI: https://doi.org/10.7554/eLife.34389.014

The following figure supplement is available for figure 4:

**Figure supplement 1.** Sequential cytosolic assembly of outer arm dynein components occurs during mammalian motile ciliogenesis in a process requiring ZMYND10.

*Figure 4 continued on next page*

*Figure 4 continued*

DOI: https://doi.org/10.7554/eLife.34389.015

stage of the chaperone cycle when ATP is hydrolysed and client and co-chaperones are released from HSP90 (*Li and Buchner, 2013*). As LRRC6 contains a p23/CS-like domain and was previously reported to interact with ZMYND10 (*Moore et al., 2013*; *Zariwala et al., 2013*), we went on to investigate the effect of LRRC6 on ZMYND10 binding to FKBP8.

Expression of myc-LRRC6 and ZMYND10-turboGFP followed by affinity purification with a monoclonal antibody to turboGFP shows that all ZMYND10 variants tested interact with LRRC6, extending on previous studies (*Zariwala et al., 2013*). Interestingly, the presence of myc-LRRC6 significantly decreased the association of ZMYND10 to FKBP8 (*Figure 8A*). These results are consistent with the canonical roles of p23 co-chaperones and suggest that its presence advances the HSP90 chaperone cycle releasing all co-chaperones and thus promoting FKBP8 dissociation from ZMYND10. Whilst we have been unable to confirm endogenous interaction of ZMYND10 and LRRC6 (*Figure 6A and B*, *Supplementary file 2*), we note that LRRC6 levels are decreased ($-7.2$ fold change, p=0.023, *Figure 5D*, *Supplementary file 1*) in *Zmynd10* mutant testes suggesting some functional interaction exists during cytoplasmic pre-assembly, highlighting the transient nature of some of these interactions in vivo.

Taken together our results suggest that ZMYND10, firstly with FKBP8 and subsequently with LRRC6, likely participates in an HSP90 chaperone cycle of common clients, the axonemal dynein heavy chains. LRRC6 may promote the final maturation and release of clients and co-chaperones onto subsequent chaperone complexes, including the R2TP and R2TP-like complexes associated with additional subunits such as the IC1/2 heterodimers, for the next stages of assembly (*Figure 8B*). Future studies will determine how HSP90 may be targeted to specific stages of dynein pre-assembly, but competitive binding through the conserved C-terminal motif MEEVD of HSP90 between discrete HSP90-co-chaperone-client complexes may confer directionality (*Back et al., 2013*; *Blundell et al., 2017*; *zur Lage et al., 2018*; *Martino et al., 2018*; *Yamaguchi et al., 2018*). Stalling of the folding process will likely trigger degradation of all folding intermediates regardless of the initial client protein recruited by the chaperone machinery.

## Discussion

Motile cilia are highly complex structures comprising of hundreds of mega-Dalton scale molecular assemblies. Axonemal dynein motors represent the largest and most complex of such motile ciliary components. Their coordinated transcription, translation, assembly and transport is critically linked to ciliary function. The cell appears to have evolved a dedicated chaperone relay system involving multiple assembly and transport factors to execute distinct steps for their pre-assembly, sometimes in a tissue- or ciliary domain-specific manner (*Dougherty et al., 2016*; *Fliegauf et al., 2005*; *Yamaguchi et al., 2018*).

In the present study, we reveal that the DNAAF ZMYND10 co-operates with the ubiquitous co-chaperone FKBP8 and chaperone HSP90 to mediate a key step in the pre-assembly pathway, specifically maturation of axonemal dynein heavy chains. Using multiple motile ciliated tissues, including trachea, oviduct and testes, from *Zmynd10* null mouse models, we observed reduced protein abundances for ODA HCs (DNAH5, DNAH9) and ICs (DNAI1 and DNAI2). Proximity ligation assays and immunopurification of endogenous components suggest that unstable intermediates of axonemal dynein ODA HCs are primarily affected and unable to fully associate with the IC heterodimers which are subsequently degraded in *Zmynd10* mutants. We provide evidence that the PPIase activity of FKBP8 is required to stabilize wild type axonemal dynein assemblies. Specific pharmacological inhibition of FKBP8 PPIase activity phenocopies the motility defects observed in *Zmynd10* mutants. We also confirm that LRRC6 may participate transiently at possibly later stages in the ZMYND10-dependent HSP90 chaperone cycle. Finally, mutations of *ZMYND10* that impair its ability to interact with the FKBP8-HSP90 chaperone system but not with LRRC6, provide a molecular explanation for a previously unresolved PCD disease-causing variant (*Zariwala et al., 2013*).

HSP90 is emerging as a central feature in the post-translational maturation of axonemal dynein motors, through at least two distinct regulatory modules. The first one involves the multi-functional

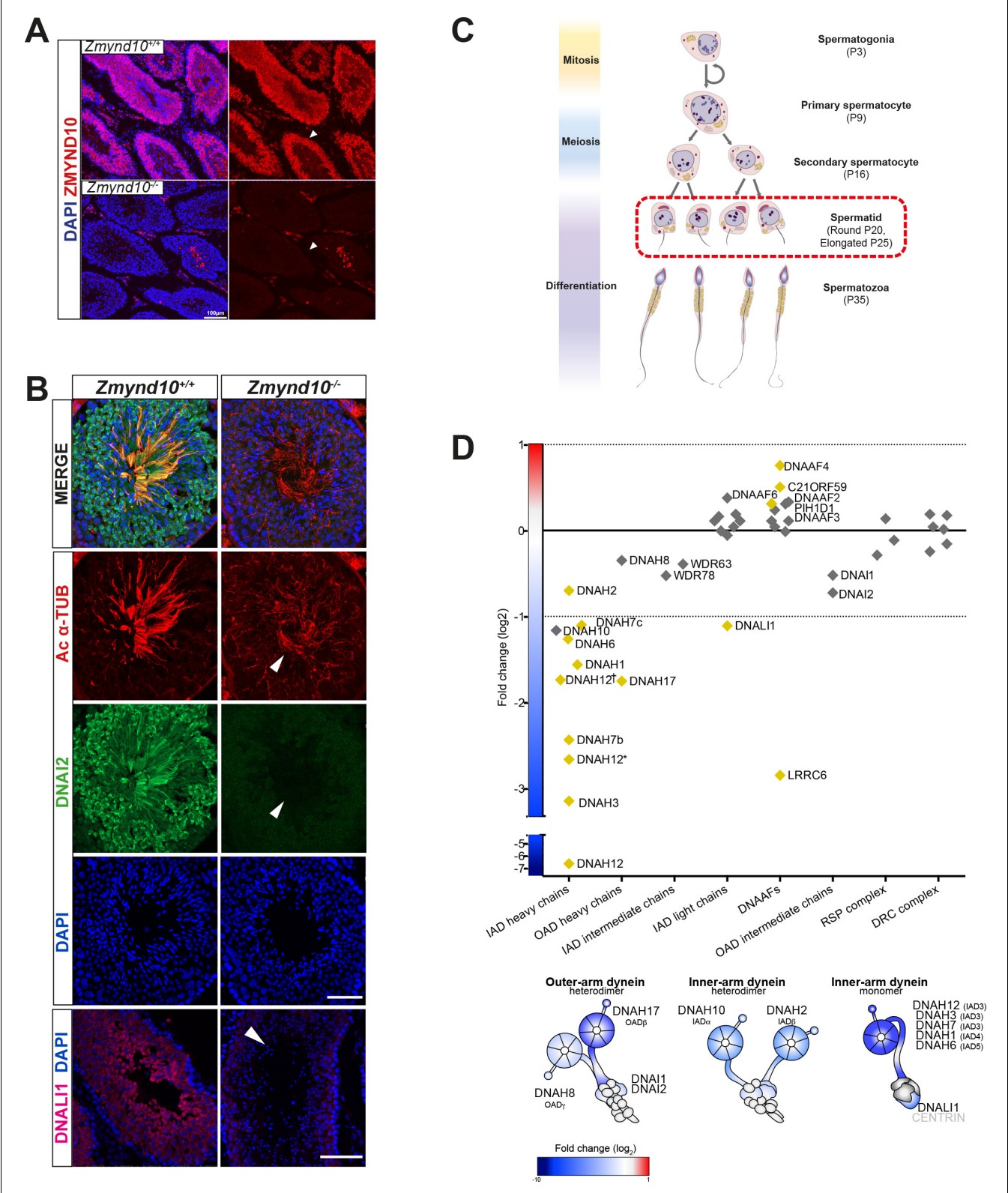

**Figure 5.** Early and specific defects in axonemal dynein heavy chain stability are observed in *Zmynd10* mutants during cytoplasmic assembly. (**A**) Immunofluorescence of ZMYND10 in control and mutant adult testes (P150) in asynchronous seminiferous tubules (arrowhead). (**A**) ZMYND10 is strongly expressed in primary spermatocytes and spermatids, where it is restricted to the cytoplasm and never in developing sperm tails. This staining is lost in mutants. Nuclei are stained with DAPI. Scale bar = 100 μm. (**B**) Cross-sections of similarly staged seminiferous tubules reveal similar developmental

*Figure 5 continued on next page*

*Figure 5 continued*

staging of sperm between adult control and mutants, but loss of DNAI2 (ODA) and DNALI1 (IDA) proteins from cytoplasm and axonemes (arrowhead) of mutant sperm. Nuclei are stained with DAPI. Scale bars = 50, 100 μm. (**C**) Schematic summarizing mouse spermatogenesis which is initially synchronized postnatally (stages shown to right), before occurring in asynchronous waves across seminiferous tubules. Axonemal dynein pre-assembly in the cytoplasm occurs from spermatid stage around P25 and continues till flagellogenesis at P30. (**D**) Unbiased quantitative proteomics of control and mutant testes (elongated spermatid: P25) reveal that loss of ZMYND10 leads to a primary reduction in abundance of all dynein HC subunits during cytoplasmic pre-assembly, whilst other components remain initially unaffected at this stage. Yellow diamonds highlight significantly ($p < 0.05$) changed hits based on LFQ intensity between *Zmynd10* mutants and wild type littermates (n = 3/genotype). Schematic below highlights fold change of specific subunits on given dynein arms. Some heavy chains had specific multiple isoforms detected (DNAH7b/c and DNAH12 (E9QPU2*, F6QA95, Q3V0Q1†). See also *Supplementary file 1*. Key: IAD: inner arm dynein; OAD: outer arm dynein; DNAAFs: dynein axonemal assembly factors; RSP: radial spoke protein; DRC: dynein regulatory complex.
DOI: https://doi.org/10.7554/eLife.34389.016

The following figure supplement is available for figure 5:

**Figure supplement 1.** Peptide intensities across the axonemal dynein heavy chains reveal profound and significant decreases, whilst other dynein subunits are not affected.
DOI: https://doi.org/10.7554/eLife.34389.017

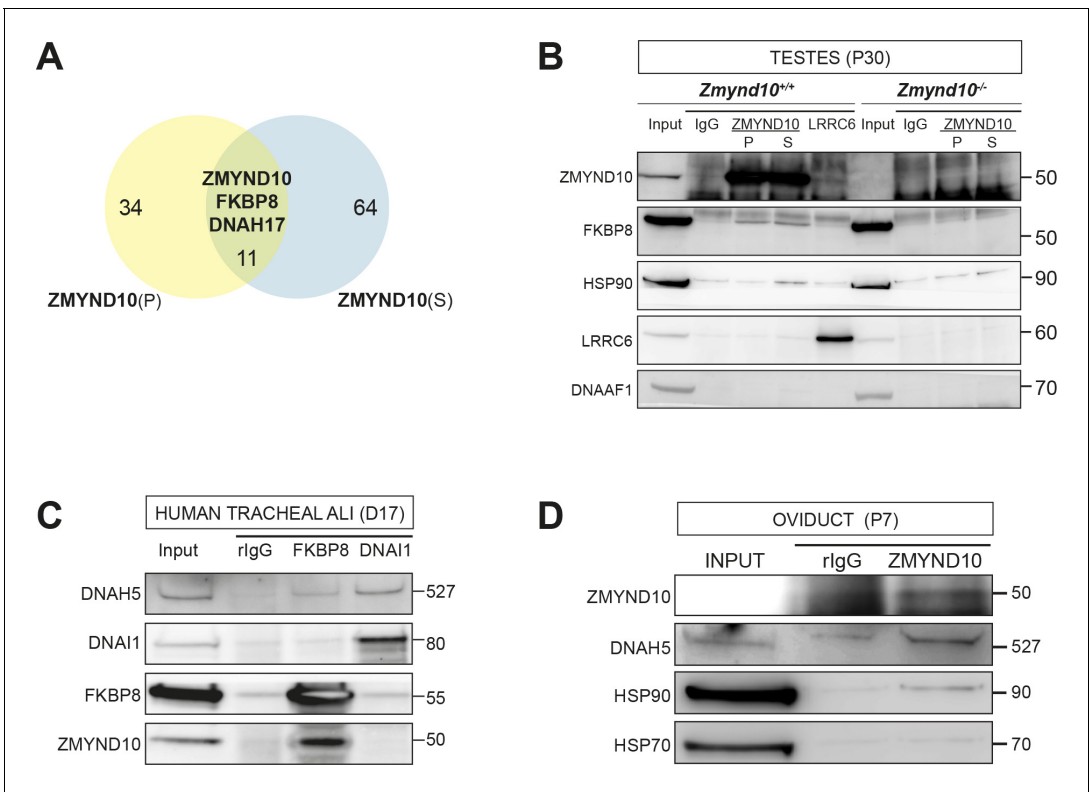

**Figure 6.** ZMYND10 interacts with a novel chaperone relay at a distinct stage of dynein heavy chain stability during cytoplasmic assembly.  (**A**) Summary schematic of affinity purification-mass spectrometry (AP-MS) analysis of endogenous ZMYND10 interactomes from P30 mouse testes, overlapping between two commercial polyclonal Sigma (S) and Proteintech (P) antibodies. See also *Supplementary file 2*. (**B**) Endogenous ZMYND10 affinity purification with two validated ZMYND10 antibodies from *Zmynd10* control and mutant P30 testes extracts confirm the ZMYND10 interaction with FKBP8 and HSP90AB1 (S only) in control samples. These interactions are not found in mutant samples lacking ZMYND10, serving as specificity controls. An in vivo interaction between ZMYND10 and LRRC6, as well as with DNAAF1, was not detected in reciprocal endogenous immunoprecipitations in P30 control testes extracts, using rIgG as a control. (**C**) Endogenous FKBP8 and DNAI1 immunoprecipitations in differentiating healthy human donor tracheal epithelial cultures (D17 ALI) both show binding of client DNAH5, the rest of the complexes are distinct, suggesting they act at sequential steps of assembly. (**D**) Endogenous ZMYND10 immunoprecipitation of client DNAH5 and chaperone HSP90, but not HSP70 from differentiating oviduct epithelial tissue (P7) using rIgG as a control. Protein molecular weights displayed in KDa.
DOI: https://doi.org/10.7554/eLife.34389.018

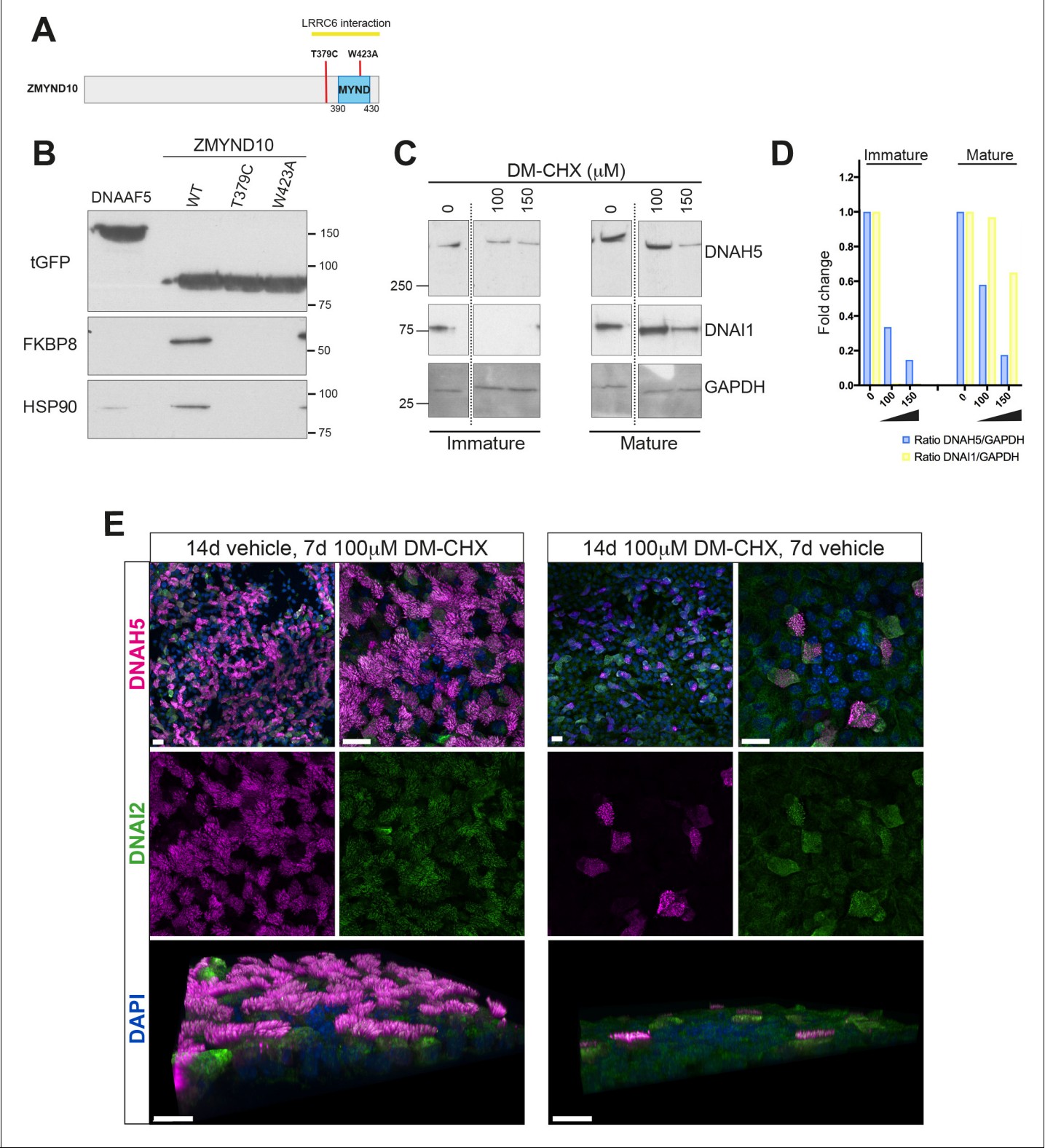

**Figure 7.** Ubiquitous FKBP8 actively participates in axonemal dynein heavy chain stability via its interaction with ZMYND10. (A) Summary of mutations generated in ZMYND10-tGFP by site-directed mutagenesis to disrupt the binding interface for FKBP8, including the W423A mutation predicted to functionally disrupt one of two $Zn^{2+}$-fingers in the MYND domain and the T379C PCD patient mutation (*Zariwala et al., 2013*), lying just before the MYND domain. (B) Extracts from transiently transfected HEK293 cells affinity purified against turboGFP shows C-terminal mutations interrupt endogenous FKBP8 and HSP90 binding to ZMYND10. (C) Healthy human donor tracheal epithelial cultures (MucilAir) before ciliation (D17, post-ALI) or
*Figure 7 continued on next page*

Figure 7 continued

fully ciliated (D60, post-ALI), were cultured for 24 hr in control (vehicle only) or DM-CHX (concentrations indicated in µM) before harvesting protein extracts. Immature cultures (D17) were more sensitive to effects of specific PPIase inhibitor DM-CHX, destabilizing dynein components, whilst mature cultures were minimally affected. (D) Quantification of band intensities for DNAH5 (blue) or DNAI1 (yellow) from (C) were normalized to loading control, and plotted as a fold change after 24 hr. (E) Z projections of whole-mount immunofluorescence of mouse tracheal epithelial cultures (MTECs) stained with DNAH5 (magenta), DNAI2 (green) and DAPI (blue) after treatment of three days post-airlift (ALI) with either 100 µm DM-CHX or vehicle control for 14 days in culture to differentiate, followed by switching cultures from control to 100 µm DM-CHX (mature: cilial beat visualized) or from DM-CHX into vehicle control (no cilia beat) and culturing an additional 7 days. Treatment of mature ciliated cultures had little effect on DNAH5 and DNAI2 levels, whilst treatment during differentiation disrupted expression of dynein subunits, as evidenced by lack of axonemal staining in most cells (right panels) but release allows recovery of dynein pre-assembly, mostly cytoplasmic after 7 days.

DOI: https://doi.org/10.7554/eLife.34389.019

R2TP complex (RUVBL1, RUVBL2, RPAP3, PIH1D1), which is a well-characterized co-chaperone of HSP90 in the assembly of several multimeric protein complexes (*Kakihara and Houry, 2012*; *von Morgen et al., 2015*). Recent reports have highlighted the importance of PIH-domain containing DNAAFs (DNAAF2 and DNAAF6), RPAP-domain containing SPAG1 as well as Reptin (RUVBL2) and Pontin (RUVBL1) as regulators of cilia motility (*Desai et al., 2018*; *Dong et al., 2014*; *zur Lage et al., 2018*; *Li et al., 2017*; *Olcese et al., 2017*; *Omran et al., 2008*; *Paff et al., 2017*;

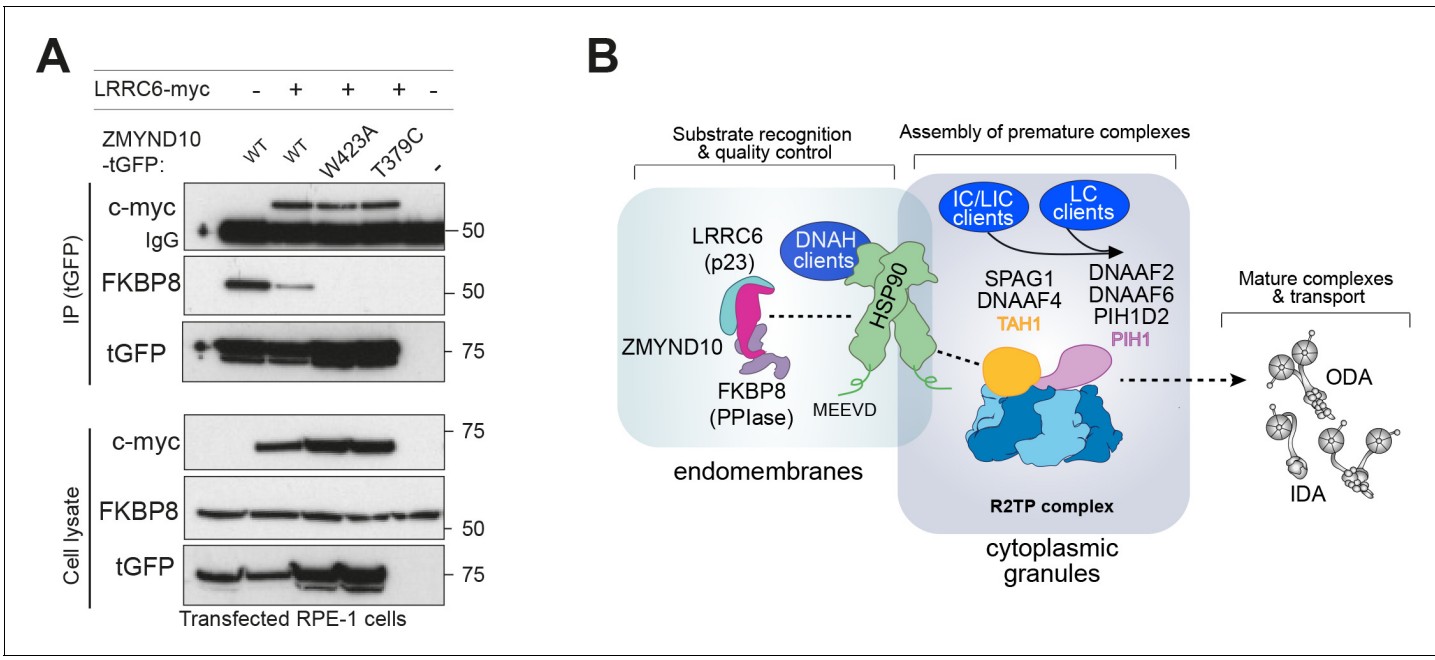

**Figure 8.** ZMYND10 specifies dynein heavy chains as clients in a chaperone relay during dynein pre-assembly. (A) Extracts from transiently transfected RPE-1 cells with ZMYND10-turboGFP variants and human LRRC6-myc were affinity purified against turboGFP (upper panels). Inclusion of LRRC6 destabilizes binding of wild type ZMYND10 to endogenous FKBP8. C-terminal ZMYND10 mutations do not affect its interaction with LRRC6. Expression of interactors is confirmed in input cell lysates (lower panels). (B) During dynein arm assembly in the cytoplasm, ZMYND10 interaction with co-chaperone PPIase FKBP8 and HSP90 is required for the stabilization and folding of dynein heavy chains (DNAH). Competitive binding of LRRC6 to ZMYND10 may advance the HSP90-FKBP8 chaperone cycle taking the client DNAHs to the next stage of assembly. We propose either R2TP or a specialized R2TP-like complex may function in parallel to assemble, scaffold or tether the DNAI1-DNAI2 heterodimeric complex via the PIH-domain proteins, such as DNAAF6, DNAAF2 and PIH1D2. The TPR-domain containing SPAG1 or DNAAF4 could recruit HSP90 via its MEEVD domain and together with RUVBL1/RUVBL2 these assembly factors could form a stable platform to promote HC-IC subunit interactions. This R2TP or R2TP-like complex likely operates distinctly to ZMYND10 as DNAI1/2 heterodimers are detected in *Zmynd10* mutants, however, these are likely degraded if fully functional mature complexes (with heavy chains) cannot be assembled. This chaperone-relay system comprises several discrete chaperone complexes overseeing the folding and stability of discrete dynein subunits. Folding intermediates are handed off to successive complexes to promote stable interactions between subunits all the while preventing spurious interactions. Stable dynein complexes, once formed are targeted to cilia via transport adaptors and intraflagellar transport (IFT).

DOI: https://doi.org/10.7554/eLife.34389.020

*Tarkar et al., 2013*; *Yamaguchi et al., 2018*). All of these factors bear homology to components or form part of the multi-functional R2TP complex. Functional genetic analysis of the four PIH zebrafish paralogues revealed distinct and overlapping DNAAF functions in pre-assembly of specific subsets of axonemal dynein motors, in a tissue-specific as well as axonemal domain-specific manner (*Yamaguchi et al., 2018*). Biochemical evidence from endogenous affinity purification of an R2TP adaptor, WDR92 further revealed physical associations of heavy and intermediate chains of ODA and IDA subunits with R2TP (*zur Lage et al., 2018*). Collectively, these studies link dynein assembly factors to a cilial-specific configuration of the HSP90-R2TP complex. However, further biochemical studies are needed to define how these different cilial-configurations of R2TP-HSP90 and dynein clients interact during the chaperone cycles operative during axonemal dynein pre-assembly. Our study uncovers a second cilial-specific configuration of the HSP90 chaperone machinery involving FKBP8 via ZMYND10 in the processing of client axonemal dynein heavy chains in all motile ciliated tissues studied (*Figure 8B*). We show that the unchanged transcript levels do not correlate with the reduced protein levels of axonemal dynein HCs (*Figure 2D*) and that unstable heavy chain folding intermediates in *Zmynd10* mutant extracts likely result from degradation of fully translated, improperly folded axonemal dynein HCs. Our results suggest that ZMYND10 directs and drives an FKBP8-HSP90 cycle, likely involving LRRC6, aiding the maturation of axonemal dynein HC clients required for their post-translational stability.

Our multipronged approach using immunofluorescence, proteomic and endogenous interaction studies all indicate that axonemal dynein HCs are the primary clients of ZMYND10. This is in contrast to recent studies suggesting that ZMYND10's primary targets are the ODA ICs, DNAI1 and DNAI2 (*Cho et al., 2018*; *Kurkowiak et al., 2016*). Initially, neither the levels of DNAI1 and DNAI2 (*Figure 5D*, *Supplementary file 2*), nor heterodimer formation (*Figure 4C*) are significantly affected during cytoplasmic pre-assembly in *Zmynd10* mutants. Moreover, our unbiased label-free quantitative proteomics shows that ZMYND10 loss also specifically impacts IDA HC stability whilst other subunits or structures remain mostly unaffected. This is distinct from the response seen in PCD models specifically affecting ODAs wherein complete lack of or misfolding of a single heavy chain in the case of DNAH5 results in a very specific and limited loss of outer dynein arms only: in *DNAH5* patients, DNALI1 is still found in the ciliary axonemes and IDAs visible by TEM (*Loges et al., 2008*). The primary defects we observe support the hypothesis that aberrant HC-IC subunit association and/or the misfolded HC polypeptides themselves trigger a robust proteostatic response leading to clearance of non-functional ODA and IDA complexes to mitigate cellular protein stress.

Our study goes towards addressing a long-standing question of where dynein pre-assembly occurs within the cytoplasm. Our interaction and mutational studies define a novel ZMYND10-FKBP8-HSP90 complex functioning in dynein pre-assembly, where FKBP8 could tether HC folding to endomembranes. The participation of FKBP8-HSP90 chaperone complex in protein folding, activation and clearance has been extensively documented (*Banasavadi-Siddegowda et al., 2011*; *Barth et al., 2009*, *2007*; *Edlich et al., 2005*; *Hutt et al., 2012*; *Saita et al., 2014*; *Taipale et al., 2014*; *Wang et al., 2006*). Given the role of this complex in folding and maturation of CFTR on the cytosolic face of the ER (*Hutt et al., 2012*; *Wang et al., 2006*), it raises the possibility that during early stages of pre-assembly, axonemal dynein heavy chains could also be localized to the cytosolic face of the ER through ZMYND10's association with FKBP8. Future work should be directed to further pinpoint where different assembly steps occur within the cytoplasm. It would be tempting to speculate that properly folded dynein HCs are subsequently exchanged in the recently described dynamic cytoplasmic puncta containing rapidly fluxing DNAAFs and less mobile dynein subunits as shown for DNAI2 and DNALI1, structures termed dynein assembly particles or DynAPs (*Huizar et al., 2017*; *Li et al., 2017*). This 'phase-separated organelle' model is attractive as increasing local concentration of specific DNAAF complexes could help overcome the apparently low binding affinities between different DNAAFs except in these molecular condensates. Indeed, we have been unable to capture endogenous interactions between LRRC6 and ZMYND10 in vivo (*Figure 6A and B*, [*Moore et al., 2013*; *Zariwala et al., 2013*]), suggesting their physiological interactions may be highly transient or temporally restricted, as opposed to existing in stable complexes.

Taken together, we propose a revised model of the dynein preassembly pathway (*Figure 8B*), where multiple roles for HSP90 are emerging. Here, ZMYND10 acts as a novel co-chaperone of the ubiquitous FKBP8-HSP90 chaperone complex for axonemal dynein HC subunit maturation. Mature, assembly-competent HCs are then handed-off to a subsequent chaperone complex, likely the R2TP

complex to allow for stable associations with other subunits such as the IC1/2 complex, in a ZMYND10-independent step. Working together, this chaperone relay ensures efficient assembly of functional dynein complexes for subsequent ciliary targeting. Given the critical role ZMYND10 plays in dynein assembly, we propose a novel alias, DNAAF7 for ZMYND10. Our work on ZMYND10 shows that the biosynthesis and quality control of dynein motors relies on an elaborate proteostasis network. Perturbations to this network by pharmacological means or due to genetic defects can disrupt motor assembly leading to PCD. This represents a paradigm shift in our understanding of PCD pathogenesis. We propose that the motile ciliopathy primary ciliary dyskinesia (PCD), when caused by defects in dynein preassembly should be considered a cell-type specific protein misfolding disease, which may be amenable to therapy by modulation of the cellular proteostasis network.

# Materials and methods

**Key resources table**

| Reagent type (species) or resource | Designation | Source or reference | Identifiers | Additional information |
|---|---|---|---|---|
| Gene (*Mus musculus*) | *Zmynd10* | NA | MGI:2387863; ENSMUSG00000010044; | Synonym: *Blu; Dnaaf7* |
| Strain (*M. musculus*) | C57BL/6J | JAX | 664 | |
| Strain (*M. musculus*) | C3H/HeJ | JAX | 659 | |
| Strain (*M. musculus*) | CD1 (ICR) | Charles River | 022 | Outbred background |
| Genetic reagent (*M. musculus*) | *Zmynd10*[em1Pmi] | This paper | Allele symbol: *Zmynd10*[em1Pmi]; Allele synonym: *Zmynd10*−; Accession ID: MGI:6159883 | CRISPR null allele of *Zmynd10*; *Zmynd10* c.695_701 p.Met178Ilefs*183 |
| Cell line (*H. sapiens*) | HEK293 | ATCC | CRL-1573 | Human embryonic kidney cell line. |
| Cell line (*H. sapiens*) | RPE-1 | ATCC | CRL-4000 | Human retinal pigmented epithelial cell line immortalized with hTERT. |
| Biological sample (*M. musculus*) | mouse tracheal epithelial cells (mTECs) | This paper | NA | See **Vladar and Brody, 2013** for protocol. |
| Biological sample (*H. sapiens*) | MucilAir tracheal epithelial cell cultures | Epithelix Sarl | EP01MD | |
| Antibody | Acetylated α-tubulin | Sigma | 6-11B-1; T6793, RRID:AB_477585 | IF (1:500–2000) |
| Antibody | β-actin | Sigma | AC-15; A1978, RRID:AB_476692 | WB (1:1000) |
| Antibody | DNAAF1/LRRC50 | Novus Biologicals | NBP2-01936; RRID: AB_2732031 | WB (1:5000) |
| Antibody | DNAH5 | PMID: 23525783 | Custom made | IF (1:100), PLA; WB (1:5000) |
| Antibody | DNAH5 | Sigma | HPA037470, RRID:AB_10672348 | IF (1:100), PLA; WB (1:5000) |
| Antibody | DNAH9 | PMID: 24421334 | Custom made | IF (1:100), PLA; WB (1:5000) |
| Antibody | DNAH5 | Sigma | HPA037470, RRID:AB_10672348 | WB (1:5000) |
| Antibody | DNAI1 | Abcam | ab171964; RRID: AB_2732030 | WB (1:5000) |
| Antibody | DNAI2 | Abnova | M01 clone IC8; H00064446-M01, RRID:AB_426059 | IF (1:100), PLA; WB (1:5000) |
| Antibody | DNAI2 | Proteintech | 17533–1-AP; 17533–1-AP, RRID:AB_2096670 | IF (1:100); WB (1:5000); IP (1.5 μg-3μg/IP) |

*Continued on next page*

*Continued*

| Reagent type (species) or resource | Designation | Source or reference | Identifiers | Additional information |
|---|---|---|---|---|
| Antibody | DNALI1 | Santa Cruz | N-13; sc-160296, RRID:AB_2246230 | IF (1:75); WB (1:1000) |
| Antibody | FKBP8 | Proteintech | 11173–1-AP, RRID:AB_10597097 | WB (1:5000); IP (1.5 µg-3µg/IP) |
| Antibody | FKBP8 | R and D Systems | MAB3580, RRID:AB_2262675 | WB (1:5000) |
| Antibody | γ tubulin | Abcam | GTU-88; ab11316, RRID:AB_297920 | IF (1:500) |
| Antibody | GAPDH | Abcam | ab8245, RRID:AB_2107448 | WB (1:5000) |
| Antibody | tGFP | Origene | TA150041, RRID:AB_2622256 | IF (1:200); WB (1:5000) |
| Antibody | GFP | Santa Cruz | FL; sc-8334, RRID:AB_641123 | WB (1:5000); IP (1.5 µg-3µg/IP) |
| Antibody | GRP-94/HSP90B1 | Thermo Scientific | clone 9G10; MA3-016, RRID:AB_2248666 | IF (1:100); WB (1:5000) |
| Antibody | HSP70 | Santa Cruz | K-20; sc-1060, RRID:AB_631685 | WB (1:5000) |
| Antibody | HSP90AB1 | R and D Systems | MAB32861, RRID:AB_2121071 | WB (1:5000) |
| Antibody | HSP90 | Santa Cruz | Clone F-8; sc-13119, RRID:AB_675659 | WB (1:5000) |
| Antibody | LRRC6 (Hiroshi Hamada) | PMID:27353389 | Custom made | WB (1:5000), a gift from Hiroshi Hamada |
| Antibody | SENTAN | Sigma | HPA043322 , RRID: AB_10793945 | IF (1:150) |
| Antibody | ZMYND10 | Proteintech | 14431–1-AP, RRID:AB_2218002 | WB (1:5000); IF (1:100); IP (1.5 µg-3µg/IP) |
| Antibody | ZMYND10 | Sigma | HPA035255, RRID:AB_10601928 | WB (1:5000); IF (1:100); IP (1.5 µg-3µg/IP) |
| Recombinant DNA reagent | pCMV6-Zmynd10-tGFP | Origene | MG207003 | Mouse *Zmynd10* ORF with C-terminal turbo-GFP tag under CMV promoter in plasmid with ampicillin resistance gene |
| Recombinant DNA reagent | pCMV6-DNAAF5-tGFP | Origene | MR221395 | Mouse *Dnaaf5* ORF with C-terminal turbo-GFP tag under CMV promoter in plasmid with ampicillin resistance gene |
| Recombinant DNA reagent | pRK5-Myc-LRRC6 | PMID:23891469 | NA | Human *LRRC6* ORF with myc tag; gift from the Hildebrandt and Gee labs |
| Recombinant DNA reagent | pX330-U6-Chimeric _BB-CBh-hSpCas9 | PMID: 23287718 | Addgene:#42230 | A human codon-optimized SpCas9 and chimeric guide RNA expression plasmid. pX330-U6-Chimeric_BB-CBh-hSpCas9 was a gift from Feng Zhang. |
| Recombinant DNA reagent | pCAG-EGxxFP | PMID: 24284873 | Addgene:#50716 | 5' and 3' EGFP fragments that shares 482 bp were placed under ubiquitous CAG promoter. Used for validation of gRNA sequences by DSB mediated EGFP reconstitution. pCAG-EGxxFP was a gift from Masahito Ikawa |

*Continued on next page*

*Continued*

| Reagent type (species) or resource | Designation | Source or reference | Identifiers | Additional information |
|---|---|---|---|---|
| Sequence-based reagent | mouse *Dnahc5* qRT-PCR primers | This paper | | AAGCTGTTGCACCAGACCAT/ CCCAGGTGGCAGTTCTGTAG; Probe:88 |
| Sequence-based reagent | mouse *Dnali1* qRT-PCR primers | This paper | | AGTTCCTGAAACGGACCAAC/ TGAGACCCATGTGGAAATGA; Probe:97 |
| Ssequence-based reagent | mouse *Zmynd10* qRT-PCR primers | This paper | | GCCATCCTTGATGCAACTATC/ CAATCAGCTCCTCCACCAG; Probe:64 |
| Sequence-based reagent | mouse *Tbp* qRT-PCR primers | This paper | | GGGGAGCTGTGATGTGAAGT/ CCAGGAAATAATTCTGGCTCA; Probe:97 |
| Chemical compound, drug | N-(N′N′-Dimethyl carboxamidomethyl) cycloheximide (DM-CHX) | PMID:16547004 | | FKBP8 inhibitor, 1 mM stock in sterile PBS |
| Software, algorithm | Fiji | PMID: 22743772 | | |
| Software, algorithm | Nis-Elements AR V4.6 | Nikon Instruments | | |
| Software, algorithm | Imaris V9.1 | Bitplane | | |
| Software, algorithm | MaxQuant | PMID: 19029910 | | |
| Software, algorithm | Andromeda | PMID: 21254760 | | |
| Software, algorithm | Perseus | PMID: 27348712 | | |
| Software, algorithm | Crapome | PMID: 23921808 | | |

## Generation of CRISPR mouse mutants

CAS9-mediated gene editing was used to generate mutant mice for *Zmynd10* (ENSEMBL: ENSMUSG00000010044) using three (guide) gRNAs each targeting 'critical' exon 6. Guide RNA sequences were cloned into a pX330 vector (Addgene:#42230) (*Cong et al., 2013*) and efficacy was first validated using a split GFP assay in HEK293 cells (Addgene: #50716) (*Mashiko et al., 2014*). Pronuclear injections of 5 ng/µl of purified plasmid DNA of pX330 constructs were injected into fertilized C57BL/6J eggs, which were cultured overnight until the two-cell stage before transferring to pseudopregnant females. PCR based screening, Sanger sequencing and characterization of genetic mutations of founder animals (F0) was performed. A genotyping was developed using a restriction digest of a PCR product for the −7 bp deletion line used in this study. Animals were maintained in SPF environment and studies carried out under the guidance issued by the Medical Research Council in 'Responsibility in the Use of Animals in Medical Research' (July 1993) and licensed by the Home Office under the Animals (Scientific Procedures) Act 1986.

## Cytology, Histology and TEM

Motile multiciliated ependymal cells were obtained from mouse brains (>P7) using a published protocol (*Grondona et al., 2013*). Mouse respiratory epithelial cells were obtained by exposing the nasal septum and scraping cells off the epithelium with an interdental brush (TePe, 0.8 mm Extra-Soft) followed by resuspension in DMEM (isolated from animals at P7-P29). Cells were spread on superfrost slides, air-dried and processed for immunofluorescence. This was modified for proximity ligation assay (PLA) where cells were resuspended in PBS, then fixed 4%PFA/3.7% sucrose/PBS for 30 min on ice and cytospun onto Superfrost slides. Human respiratory epithelial cells obtained by brush biopsying the nasal epithelium of healthy human donors or P7 neonatal mice were processed for proximity ligation assay using a Duolink PLA starter kit (DUO92101, Sigma-Aldrich), as per the manufacturer's instructions following PFA fixation and 0.25%Triton-X100/TBS permeabilization 10 min. Alexa-488 phalloidin (Thermo Fischer) or rat anti-GRP94 (Thermo Fischer) counterstaining was done post-PLA protocol, prior to mounting in Duolink In Situ Mounting Medium with DAPI (Sigma

Aldrich). Trachea (P7), testes (P150) and oviducts (P7) were dissected and immersion fixed in 4% paraformaldehyde (from 16% solution, Thermo Fischer) overnight and cryosectioned for immunofluorescence staining with antibodies to ZMYND10, acetylated α-tubulin or dynein components (*Diggle et al., 2014*). Nasal turbinates were similarly fixed and processed for paraffin sectioning stained with H and E to reveal mucus plugs. Immunofluorescence images were acquired at either 60x or 100x optical magnification as confocal stacks through whole cells and tissue sections using a Nikon A1R confocal microscope and displayed as Z-projections. Epidydymal spermatozoa were isolated by dissecting the cauda and caput regions of the epididymides in M2 media (Life Technologies), spread onto superfrost slides and air-dried followed by fixation and permeabilisation for immunofluorescence, as previously described (*Diggle et al., 2014*). For counting, sperm from the cauda epididymides were immobilized by diluting in $H_2O$ and counts were performed using a haemocytometer. For transmission electron microscopy, trachea tissue samples were dissected into PBS and immersion fixed in 2% PFA/2.5% glutaraldehyde (Sigma-Aldrich)/0.2M Sodium Cacodylate Buffer pH7.4 with 0.04% $CaCl_2$ (*Hall et al., 2013*). Samples were cut into semi-thin and ultrathin sections and imaged by transmission electron microscopy (EM Services, Newcastle University Medical School).

## Live brain sectioning and high-speed videomicroscopy of ependymal cilia

Whole brains were isolated from neonatal mice in ice cold PBS and kept on ice. Brains were mounted vertically along the caudo-rostral axis on a petri dish and embedded in low melting point agarose (Thermo Scientific). 400 µm thick vibratome sections of live brain tissue were obtained and floated onto wells of a glass bottom multiwell plate (Greiner Sensoplates cat.662892) containing DMEM and maintained at 37°C and 5% $CO_2$. Sections were imaged on a Nikon macroscope to visualize dilated lateral ventricles. Motile cilia beating along the surfaces of the lateral walls were visualized and motility was recorded using a high-speed videomicroscopy Andor CCD camera attached to a confocal capture set-up.

## Immunoprecipitations (IP) and immunoblots

Endogenous immunoprecipitations were performed using protein extracts either from multiciliated cell cultures or motile ciliated tissues lysed under mild lysis conditions (50 mM Tris-HCl (pH 7.5), 100 mM NaCl, 10% Glycerol, 0.5 mM EDTA, 0.5% IGEPAL, 0.15% Triton-X 100 and Halt Protease Inhibitor Single use cocktail EDTA free (Thermo Fischer)). For detecting HSP90 interactions, we included sodium molybdate (Sigma-Aldrich) in the IP buffer aiming to reduce ATP hydrolysis and client release (*Sullivan et al., 2002*). To detect interactions between ODA subunits, a DNAI2 antibody (Abnova H00064446-M01, RRID:AB_426059) was used as a bait to enrich DNAI2 containing complexes from mouse trachea and oviduct lysates. Immunoblotting was performed using DNAI1 (Abcam ab171964, RRID:AB_2732030) and DNAH5 (N-terminal epitope, M. Takeda). For ZMYND10 interaction studies, extracts from whole testes (P30) and differentiating mouse P7 oviducts were used. Endogenous ZMYND10 containing complexes were pulled out using two validated ZMYND10 polyclonal antibodies (Sigma HPA035255, RRID:AB_10601928; Proteintech 14431–1-AP, RRID:AB_2218002). Immunoblotting was performed using an HSP90 antibody (Santa Cruz sc-13119, RRID:AB_675659). For human samples, endogenous FKBP8, DNAI1 and DNAI2 pulldowns were perfomed on lysates from normal human airway epithelial cells (MucilAir, Epithelix Sarl) grown at air-liquid interface for 17 days (immature cells). Antibodies for FKBP8 (Proteintech 11173–1-AP, RRID:AB_10597097), DNAI1 (Abcam ab171964; RRID:AB_2732030) and DNAI2 (Abnova H00064446-M01, RRID:AB_426059) were used as baits and antibodies for DNAH5 (Sigma HPA037470, RRID:AB_10672348) and ZMYND10 (Proteintech 14431–1-AP, RRID:AB_2218002) were used to detect these interactors. An isotype-matched IgG rabbit polyclonal antibody (GFP: sc-8334, RRID:AB_641123, Santa Cruz) was used as control. In all pull-down experiments, immunocomplexes were concentrated onto Protein G magnetic beads (PureProteome, Millipore). Following washes, immunocomplexes were eluted off the beads by boiling and resolved by SDS-PAGE for immunoblotting. Alternatively, beads were processed for on-bead tryptic digestion and mass-spectrometric analysis. For overexpression pull-downs, mouse *Dnaaf5-tGFP* (Origene- MG221395), *Zmynd10-tGFP* (Origene, MG207003) and *myc-Lrrc6* (*Zariwala et al., 2013*) were transiently transfected (Lipofectamine2000

into hTERT-RPE (ATCC CRL-4000) and HEK293 (ATCC CRL-1573) cells, which were tested regularly for mycoplasma. Site-directed mutagenesis was performed using two complementary PCR primers containing the desired nucleotide changes (PrimerX tool) to amplify *Zmynd10-tGFP* with proof reading DNA polymerase (Agilent II), followed by DpnI digestion, E coli transformation and sequencing of the thus recovered plasmids. Subsequent affinity purification using a turboGFP antibody (Evrogen TA150041, RRID:AB_2622256) was used to isolate fusion proteins 24-hr post-transfection followed by immobilization onto protein G beads. For immunoblots, proteins were resolved by SDS-PAGE using 3–8% Tris-Acetate gels or 4–12% Bis-Tris precast gels (NuPage Life Technologies), then transferred using XCell II Blot module (Life Technologies) to either nitrocellulose or PVDF membranes followed by manual or iBind Western (Thermo Fisher) system for antibody binding. Protein bands were detected using SuperSignal West Femto or Pico kit (Thermo Scientific). *Supplementary file 3* contains a list of reagents used.

## Mass spectrometry and proteomic data analysis

For whole tissue proteome analysis, the Filter Aided Sample Preparation (FASP) method was used (*Wiśniewski et al., 2009*). Briefly, mouse testes samples were homogenized in a lysis buffer consisting of 100 mM Tris (hydroxymethyl)amino-methane hydrochloride (Tris-HCl), pH 7.5, in presence of protease (Complete Mini Protease Inhibitor Tablets, Roche and 1 mM Phenylmethylsulfonyl fluoride,, Sigma) and phosphatase inhibitors (PhosSTOP Phosphatase Inhibitor Cocktail Tablets, Roche). Samples were further processed and peptides and proteins were identified and quantified with the MaxQuant software package, and label-free quantification was performed by MaxLFQ, as described in (*Hall et al., 2017*). The false discovery rate, determined by searching a reverse database, was set at 0.01 for both peptides and proteins. All bioinformatic analyses were performed with the Perseus software. Intensity values were log-normalized, 0-values were imputed by a normal distribution 1.8 $\pi$ down of the mean and with a width of 0.2 $\pi$. Statistically significant variance between the sample groups was tested by a permutation-based FDR approach and a Student's t test with a p value cut-off of 0.01. Total proteomic data are available via ProteomeXchange with identifier PXD006849 and are summarised in *Supplementary file 1*

To examine endogenous ZMYND10 interactions from postnatal day 30 (P30: a period of synchronized flagellogenesis) testes extracts using two well-validated polyclonal antibodies (ZMYND10 Proteintech and Sigma) using an IP/MS workflow carried according to (*Turriziani et al., 2014*). Mass spectra were analysed using MaxQuant software and label-free quantification intensity values were obtained for analysis. T-test p-values between MS runs were calculated. MS datasets were ranked by $\log_2$ fold-change (enrichment) over IgG controls (*Supplementary file 2*). As a filtering strategy to find 'true' interactions, we used the CRAPome repository (http://www.crapome.org/.) containing a comprehensive list of the most abundant contaminants commonly found in AP/MS experiments (*Mellacheruvu et al., 2013*). To aid filtering, we used an arbitrary threshold of 25 (i.e. proteins appearing in >25 out of 411 experiments captured in the CRAPome repository) were removed from further analysis. Filtered interactors common to both ranked datasets were prioritized for further studies for validation as interactors of ZMYND10 in vivo. The mass spectrometry proteomics data have been deposited to the ProteomeXchange Consortium via the PRIDE partner repository with the dataset identifier PXD006849 and summarised in *Supplementary file 2*.

## Mouse tracheal epithelial cultures

Mouse tracheal epithelia cells (mTECs) were isolated from the tracheas of 5–7 week old outbred mice, and then studied as passage 0 cells (*Vladar and Brody, 2013*). Cells were cultured on semi-permeable supported membranes (Transwell; Costar, Corning, NY), as previously described (*Vladar and Brody, 2013*). Y276342 (StemCell, UK) at 10 mM was added to the medium during the proliferation stages to promote basal cell proliferation.

## Reverse transcription quantitative real time-PCR (RT qPCR)

Total RNA was isolated from freshly dissected tissue or tissue stored in RNAlater (Qiagen). Isolation was carried out using RNeasy Mini Kit or RNeasy Fibrous Tissue Mini Kit (Qiagen) following manufacturer's protocol. RNA samples were treated with Turbo DNAse to remove genomic DNA contamination using the Turbo DNA free kit (Ambion). Intron-spanning RT-qPCR assays were designed using

the Universal Probe Library probe finder tool (Roche) to identify transcript specific primer-probe sets listed in supplementary table. Three separate experimental runs were carried out for each plate. All runs were done on three individual biological replicates. Data was analysed using Roche LC480 software. Subsequently, a paired two-tailed students t-test was used to compare differences in the mean expression values between wild type and mutant samples.

## DM-CHX FKBP8 inhibitor studies

Lyophilized FKBP8 inhibitor N-(N'N'-Dimethylcarboxamidomethyl)cycloheximide (DM-CHX) (*Edlich et al., 2006*) was dissolved in sterile PBS in a 1 mM stock and diluted further to working concentrations in MucilAir media (EPITHELIX Sàrl). MucilAir tracheal epithelial cultures (EPITHELIX Sàrl) inserts from healthy human donors (same for each stage, immature D17 after air-lift or mature D60 after air-lift) were incubated with DM-CHX for 24 hr at the indicated concentrations and harvested in mild lysis buffer, (50 mM Tris-HCl (pH 7.5), 100 mM NaCl, 10% Glycerol, 0.5 mM EDTA, 0.5% IGE-PAL, 0.15% Triton-X 100 and Halt Protease Inhibitor Single use cocktail EDTA free (Thermo Fischer). For FKBP8 inhibition studies of mTECs, DM-CHX was diluted to 100 µm in NuSerum media and added to wells every 2 days from 3 days ALI (immature treatment) or after 14 days (mature treatment). For detection of DNAH5 (Takeda custom) and DNAI2 (Abnova H00064446-M01, RRID: AB_426059) from mTECs, membrane from inserts were removed and stained with antibodies as described above.

## Imaging

Fluorescent confocal images (PLA, IF, ICC) were acquired using a 60x Apochromat λS or 100x Plan Apochromat VC 1.4 DIC N2 lens using a Nikon A1R confocal microscope. Data were acquired using NIS Elements AR software (Nikon Instruments Europe, Netherlands). mTEC wholemount images were acquired using 20x Plan Apochromat VC 0.75 DIC N2 or air 40x Plan Fluar 0.75 DIC N2 lens on the multimodal Imaging Platform Dragonfly (Andor Technologies, Belfast UK). Data were collected in Spinning Disk 25 µm pinhole mode on the high sensitivity iXon888 EMCCD camera. Z stacks were collected using a Mad City Labs Piezo. Data was visualized using IMARIS 8.4 (Bitplane).

## Statistics

Statistical tests were performed using GraphPad Prism 7, (GraphPad Software, California) as described in the text.

## Acknowledgements

The authors thank IGMM technical services, the IGMM Advanced Imaging and CBS animal facilities for advice and technical assistance. We are very grateful to Masayuki Amagai, Hiroshi Hamada, Aki-haru Kubo, Yasuko Inaba, Friedhelm Hildebrandt, and Heon Yung Gee for generously sharing reagents and to Chrisostomos Prodromou for helpful discussions. This work was supported by core funding from the MRC (MC_UU_12018/26) (to GRM, PLY, DOD, PAT, MAK, PM), and a Science Foundation Ireland grant (SIRG) (to AGM and AvK). The mass spectrometry proteomics data have been deposited to the ProteomeXchange Consortium via the PRIDE partner repository with the dataset identifier PXD006849.

## Additional information

### Funding

| Funder | Grant reference number | Author |
|---|---|---|
| Medical Research Council | MRC_UU_12018/26 | Girish R Mali<br>Patricia L Yeyati<br>Daniel O Dodd<br>Peter A Tennant<br>Margaret A Keighren<br>Pleasantine Mill |

| Science Foundation Ireland | Amaya Garcia-Munoz<br>Alex von Kreisheim |
| --- | --- |
| Carnegie Trust for the Universities of Scotland | Girish R Mali<br>Andrew Paul Jarman<br>Pleasantine Mill |

The funders had no role in study design, data collection and interpretation, or the decision to submit the work for publication.

## Author contributions

Girish R Mali, Conceptualization, Formal analysis, Validation, Investigation, Writing—original draft, Writing—review and editing; Patricia L Yeyati, Formal analysis, Validation, Investigation, Writing—review and editing; Seiya Mizuno, Peter A Tennant, Margaret A Keighren, Resources, Investigation; Daniel O Dodd, Validation, Investigation, Visualization; Petra zur Lage, Conceptualization, Investigation; Amelia Shoemark, Atsuko Shimada, Hiroyuki Takeda, Frank Edlich, Satoru Takahashi, Resources, Methodology; Amaya Garcia-Munoz, Formal analysis, Methodology; Alex von Kreigsheim, Data curation, Formal analysis, Methodology; Andrew P Jarman, Conceptualization, Supervision, Writing—review and editing; Pleasantine Mill, Conceptualization, Supervision, Funding acquisition, Investigation, Writing—original draft, Project administration, Writing—review and editing

## Author ORCIDs

Girish R Mali (ID) http://orcid.org/0000-0002-5551-7555
Patricia L Yeyati (ID) http://orcid.org/0000-0001-5094-5182
Andrew P Jarman (ID) https://orcid.org/0000-0003-4036-5701
Pleasantine Mill (ID) http://orcid.org/0000-0001-5218-134X

## Ethics

Animal experimentation: All animal work was approved by a University of Edinburgh internal ethics committee and was performed in accordance with institutional guidelines under license by the UK Home Office (PPL 60/4424). Mice were maintained in an SPF environment in facilities of the University of Edinburgh (PEL 60/2605).

## Decision letter and Author response

Decision letter https://doi.org/10.7554/eLife.34389.036
Author response https://doi.org/10.7554/eLife.34389.037

# Additional files

### Supplementary files

• Source data 1. *Figures 1–8* Uncropped immunoblots Images show un-cropped versions of immunoblots. Figure panels used are demarcated with a box and the detected proteins are labeled. Asterisks are used to denote either non-specific bands (6C, 6D and 7B) or putative degradation products (7E). HE = High exposure, LE = Low exposure
DOI: https://doi.org/10.7554/eLife.34389.021

• Supplementary file 1. Summary of label-free quantitative global proteomics of P25 testes from *Zmynd10*$^{+/+}$ and *Zmynd10*$^{-/-}$ mice. Label free quantitative (LFQ) proteomics were performed on whole testes extracts to compare differential protein expression profiles between *Zmynd10*$^{-/-}$ mutants and age matched (P25) wild type littermate controls. LFQ intensities from three sets of biological replicates were averaged per genotype and ratios of enrichment or depletion in mutant versus wild type were calculated. Significance values were calculated using a Student's paired 2-tailed t-test. For ease of interpretation, the proteins were grouped into their respective complexes (i.e. Inner Dynein Arm Heavy Chains, Outer Dynein Arm Heavy Chains etc.)
DOI: https://doi.org/10.7554/eLife.34389.022

• Supplementary file 2. Endogenous ZMYND10 immunoprecipitation protein interactor profiling. Filtered by significance. Endogenous ZMYND10 was immunoprecipitated from total extracts of control

wild type testes (P30) with two commercial polyclonal antibodies (Proteintech and Sigma). Proteins that co-precipitated with ZMYND10 in triplicate experiments were identified by tandem Mass Spectrometry and ranked according to fold-change (FC Ratio) enrichment and significance (t-test p-value) in the ZMYND10 specific pull-downs versus control immunoglobulin pull-downs. Proteins that commonly precipitated with ZMYND10 using the two separate polyclonal antibodies were considered putative interactors. A ranked list of these common hits is provided with ZMYND10 and the interactors investigated in this study that is FKBP8, HSP90AB1 and DNAH17 highlighted in green.
DOI: https://doi.org/10.7554/eLife.34389.023

• Supplementary file 3. Table of antibodies, qRT-PCR primers and plasmids used in this study.
DOI: https://doi.org/10.7554/eLife.34389.024

• Supplementary file 4. Table of additional mouse primers used in this study.
DOI: https://doi.org/10.7554/eLife.34389.025

• Transparent reporting form
DOI: https://doi.org/10.7554/eLife.34389.026

## Data availability

The mass spectrometry proteomics data have been deposited and are available on the ProteomeXchange Consortium via the PRIDE partner repository with the dataset identifier PXD006849.

The following dataset was generated:

| Author(s) | Year | Dataset title | Dataset URL | Database, license, and accessibility information |
|---|---|---|---|---|
| Mali GM | 2017 | ZMYND10 functions in a chaperone-relay during axonemal dynein assmbly | http://proteomecentral. proteomexchange.org/ cgi/GetDataset?ID= PXD006849 | Publicly available at ProteomeXchange (accession no. PXD006849) |

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
