## [Decision Letter]

Thank you for submitting your article "ZMYND10 functions in a chaperone relay during axonemal dynein assembly" for consideration by *eLife*. Your article has been reviewed by three peer reviewers, one of whom is a member of our Board of Reviewing Editors, and the evaluation has been overseen by Michael Marletta as the Senior Editor. The following individual involved in review of your submission has agreed to reveal his identity: David Mitchell (Reviewer #2).

The reviewers have discussed the reviews with one another and the Reviewing Editor has drafted this decision to help you prepare a revised submission.

This manuscript describes generation of a *Zmynd10* null mouse and analysis of the effects of this mutation on axonemal dynein assembly. Both the macroscopic phenotype and the microscopy analysis show that this mutation disrupts ciliary motility through blocking assembly of axonemal dynein arms. Blots of heavy chains DNAH5 and DNAH9, as well as intermediate chains DNAI1 and DNAI2 supported IF studies of both trachea and oviduct. Maintained mRNA abundance and presence of degradation products on blots supported the conclusion that reduction of these proteins was due to proteolysis, not a block to transcription or translation.

Overall, this paper provides some solid advances in understanding mechanisms of axonemal dynein assembly and the role of ZMYND10. However, it requires more attention to detail in describing and presenting the data in order to be understood and accepted by the readership of this journal.

Essential revisions:

1) An attempt to show that proteins were first made and could be visualized in the cytoplasm, and were later either assembled (normal cells) or degraded (mutant cells) was of interest, but the data presentation made interpretation of these results problematic. The panels in Figure 3 are poorly labeled, and poorly described in the figure legend, making it difficult to follow the logic of the description of results (subsection “Mis-assembled dynein motors are blocked from entering cilia and cleared in *Zmynd10* mutants”, last paragraph). The authors suggest that they can differentiate between immature and mature cells, but use no markers. If the presence of DNAI2 in the cytoplasm is used to identify immature cells, then the reasoning is circular. Additionally, if the cells in panel B are all immature, as suggested by the arrowheads, then why is there no DNAI2 or DNALI1 signal in the cytoplasm? Second, panel B is described as showing IDA proteins, but includes one IDA protein (DNALI1) and one ODA protein (DNAI2). Further, it appears that DNAH5 is present in the cilia of an "immature" cell in the lower panel of Figure 3A. If DNAH5 at first assembles and then at a later time point is removed from the cilia, this is an important point and should be more completely documented. Lastly, if the cells indicated by arrows are mature, and the cilia are supposed to be longer, why do the cilia on the mature +/+ cells in panel A appear the shortest, and those on the immature +/+ cells in panel B appear the longest? Figure 4A suffers from the same difficulty in knowing which cells are mature and which are immature, without any independent markers. Thus, clarification of the maturity of these cells is critical for making claims about how dynein arm assembly changes over cell maturity.

2) Subsection “Mis-assembled dynein motors are blocked from entering cilia and cleared in *Zmynd10* mutants”, first paragraph and Figure 2F. In the mutant lysates, multiple (4 are indicated) fragments of DNAH5 (OAD HC) are seen. Although at lower levels, these also appear to be present in the control. Given that dynein HCs are very protease sensitive (what is likely a proteolytic fragment is clearly seen below the full-length DHC in the control lane), is it clear that this observation indicates "post-translational destabilization" rather than either enhanced proteolysis (even post-cell lysis) or premature translation termination? What was the antigen used to generate the DNAH5 antibody? Knowing this might allow the authors to conclude whether these represent N-terminal fragments that could have resulted from premature termination or C-terminal (motor domain) fragments that must have resulted from proteolysis.

3) Figure 4C and 4D are confusing. Why are different tissues compared at different time points and using different antibodies (DNAI2 and DNAI1 versus DNAI2 and DNAH5)? It is not possible to assess whether differences in assembly formation or stability are due to tissue-specific differences, maturation differences, the different proteins examined, or all of the above. Also, the diagram in Figure 4F is not consistent with the preceding data, which shows formation of complexes between the IC dimer and DNAH5, even if DNAH5 itself is unstable as indicated by the presence of degradation products.

4) Figure 6A and B: Positive controls are required to make sense of negative IP data. For example, if DNAAF2 is confirmed to pull down DNAAF4, but does not pull down ZMYND10, I would be reasonably confident that DNAAF2 does not strongly interact with ZMYND10. Zariwala et al. and Moore et al. identified a biochemical interaction between LRRC6 and ZMYND10. This work fails to recapitulate that interaction. The authors should use the same reagents as Zariwala to resolve this discrepancy. Do they interact or not (or do so extremely transiently)?

5) The authors provide very interesting data on the role of FKBP8's peptidyl prolyl isomerase activity in dynein assembly and see a decrease in both DHAH5 and DNAI1 stability when this is inhibited. Were these effects specific to these two particular outer arm subunits or is this manifest in others as well e.g. DNAI2 (which associates directly with DNAI1 and for which antibodies are available). In other words, is PPIase activity itself needed for stability of all these components individually, or is their loss the consequence of lack of PPIase activity on one key component of this massive complex?

6) Some points are overstated. The authors suggest that, "the motile ciliopathy Primary Ciliary Dyskinesia (PCD) should be considered a cell-type specific protein-misfolding disease." Surely, the authors mean that only some forms of PCD should be considered as being caused by protein misfolding.

---

## [Author Response]

Essential revisions:1) An attempt to show that proteins were first made and could be visualized in the cytoplasm, and were later either assembled (normal cells) or degraded (mutant cells) was of interest, but the data presentation made interpretation of these results problematic. The panels in Figure 3 are poorly labeled, and poorly described in the figure legend, making it difficult to follow the logic of the description of results (subsection “Mis-assembled dynein motors are blocked from entering cilia and cleared in Zmynd10 mutants”, last paragraph). The authors suggest that they can differentiate between immature and mature cells, but use no markers. If the presence of DNAI2 in the cytoplasm is used to identify immature cells, then the reasoning is circular.

We agree on the importance of clarifying this point of initial attempts at assembly, followed by clearance when this fails in mutants. Figures 3 and 4 are now better annotated and a more detailed description in the figure legends is provided. We have also used an independent marker for motile ciliary maturation, SENTAN (Kubo et al., 2008), which is enriched in the apical structure of mature motile ciliary tips. Using two commercial SENTAN antibodies on healthy human donor nasal brushings (here, new Figure 3A), we have demonstrated that the immunostaining pattern for two axonemal dynein subunits DNAI2 and DNALI1, transitions from a predominantly cytoplasmic (SENTAN negative, immature cell stage) to exclusively ciliary (SENTAN positive, mature cell stage) staining during motile ciliated cell differentiation for. We received a small aliquot of un-purified serum against mouse SENTAN (Kubo et al., 2008), which also showed enrichment in cilial tips but less precisely during differentiation (Author response image 1), as dyneins progressively translocate from the cytoplasm to cilia. We have not included the mouse data in the paper (see Author response image 1, right panel) as it requires much more optimization.

Additionally, if the cells in panel B are all immature, as suggested by the arrowheads, then why is there no DNAI2 or DNALI1 signal in the cytoplasm? Second, panel B is described as showing IDA proteins, but includes one IDA protein (DNALI1) and one ODA protein (DNAI2).

We thank the reviewer’s keen eye on our image annotation in this figure and have re-annotated these offending panels (Figure 3C), where only the mutant cell is immature, showing accumulation apically. The figure legends have also been corrected. Unfortunately, DNALI1 is the only good commercially available IAD subunit antibody, hence the MS analysis in Figure 5 to circumvent these limitations.

Further, it appears that DNAH5 is present in the cilia of an "immature" cell in the lower panel of Figure 3A. If DNAH5 at first assembles and then at a later time point is removed from the cilia, this is an important point and should be more completely documented. Lastly, if the cells indicated by arrows are mature, and the cilia are supposed to be longer, why do the cilia on the mature +/+ cells in panel A appear the shortest, and those on the immature +/+ cells in panel B appear the longest? Figure 4A suffers from the same difficulty in knowing which cells are mature and which are immature, without any independent markers. Thus, clarification of the maturity of these cells is critical for making claims about how dynein arm assembly changes over cell maturity.

We appreciate the reviewer’s keen eye but note that the perceived ‘presence’ of DNAH5 in cilia axonemes of immature mutant cells is likely due to the images being acquired with increased pixel dwell time to better visualize the complete absence of DNAH5 signal in neighboring mature mutant cells, further highlighting the robust clearance of DNAH5. We have amended this panel to clarify the absence of DNAH5 signal in mutant cilia. We reiterate that we never observe DNAH5 staining in the ciliary compartment in *Zmynd10* mutant cells. The representative images acquired at comparable imaging conditions are provided to additionally emphasize this point (Author response image 2).

**Author response image 2. respfig2:** 

2) Subsection “Mis-assembled dynein motors are blocked from entering cilia and cleared in Zmynd10 mutants”, first paragraph and Figure 2F. In the mutant lysates, multiple (4 are indicated) fragments of DNAH5 (OAD HC) are seen. Although at lower levels, these also appear to be present in the control. Given that dynein HCs are very protease sensitive (what is likely a proteolytic fragment is clearly seen below the full-length DHC in the control lane), is it clear that this observation indicates "post-translational destabilization" rather than either enhanced proteolysis (even post-cell lysis) or premature translation termination? What was the antigen used to generate the DNAH5 antibody? Knowing this might allow the authors to conclude whether these represent N-terminal fragments that could have resulted from premature termination or C-terminal (motor domain) fragments that must have resulted from proteolysis.

We think that this observation is extremely interesting, as it would inform on the mechanism of action of ZMYND10. From the position of the antigen used to raise this antibody (amino acids 846-1136; see Matsuo et al. 2013 AJP: LCMP), one can argue that the high molecular weight polypeptides detected in the immunoblots are unlikely to originate from premature translational termination. The antigen in the detected fragments maps to the stem region of the dynein heavy chain and does not include any region of the motor domain. The antigen does not lie to the extreme N-terminus of the protein and the high molecular weight fragments detected in the immunoblots appear consistent with a long (majority of) polypeptide being synthesised. To investigate this further, we analysed individual polypeptides detected for the relevant dynein proteins in the total testes proteomes and plotted their intensity along the length of the corresponding full-length proteins. Comparing the intensities of axonemal heavy chains in wild type and *Zmynd10* mutant samples indicates that identified peptides mapped all along the proteins but at lower intensity (see new Figure 5—figure supplement 1).

The presence of terminal peptides albeit with lower intensity (less abundant) in the mutants, argues against premature translational termination and suggests instead that proteins are fully synthesized and latter degraded or cleared. In agreement, the decreased DNAH5 levels in *Zmynd10* mutant fixed tissue sections indicate that its degradation is not post-cell lysis but may instead be part of a controlled mechanism that clears stalled dynein motor complexes in the absence of the assembly factor. As such, we interpret the high molecular weight laddering pattern of fragments, characteristic of proteolytic degradation, to be a consequence of loss of the ZMYND10-FKBP8-HSP90 mediated chaperoning of DHCs and therefore leading to enhanced protease sensitivity above the normal levels of sensitivity observed in control lysates.

3) Figure 4C and 4D are confusing. Why are different tissues compared at different time points and using different antibodies (DNAI2 and DNAI1 versus DNAI2 and DNAH5)? It is not possible to assess whether differences in assembly formation or stability are due to tissue-specific differences, maturation differences, the different proteins examined, or all of the above. Also, the diagram in Figure 4F is not consistent with the preceding data, which shows formation of complexes between the IC dimer and DNAH5, even if DNAH5 itself is unstable as indicated by the presence of degradation products.

Multiciliated cells appear at different developmental time points in different tissues in mammals. For instance, in mice, multiciliated cells start to appear in the trachea as early as embryonic day 15.5 (E15.5) whereas in the brain and oviducts, they appear postnatally, from P5 and P8 onwards respectively. Despite the differences in timing, all multiciliated cells must undergo the same programme of differentiation, which includes the preassembly of dyneins in the cytoplasm. We believe that the strength of our study is the use of a mammalian model that allows us to interrogate basic principles of dynein assembly in different tissues, describing a universal function for ZMYND10 in all motile ciliated lineages examined. This is in contrast to the recent study using zebrafish mutants for all 4 PIH protein paralogues, which show distinct and sometimes redundant functions in dynein assembly in a tissue-specific manner (Yamaguchi et al., 2018). In our global view of axonemal dynein assembly, we have analysed seemingly different stages (i.e. days post-natal), which reflect these tissue-specific differences in timing of when the motile multi-ciliogenic programme gets switched on. To analyse formation of the dynein intermediate chain IC heterodimer (IC1/IC2), we used P26 testes samples (cytoplasmic pre-assembly just before flagellogenesis). Here, DNAI1 is precipitated by DNAI2 at similar levels in both control and mutants suggesting this complex is not initially affected (Figure 4C). Importantly, only slight differences in the levels of DNAI1 and DNAI2 are seen in the mutant inputs (and validated in our MS data Figures 5D, Figure 5—figure supplement 1). However, we observe that in both differentiating oviducts (Figure 4D) and trachea (new Figure 4—figure supplement 1B), the ability of DNAI2 to precipitate DNAH5 is weaker in the mutants compared to controls, even when normalizing for slight reductions in inputs in mutants. This indicates that the subsequent IC-HC interaction is somehow compromised in the absence of ZMYND10 (see new Figure 4E). For technical reasons, we were unable to assess the IC-HC interaction in P26 testes, to complement this series as we have been unable to source specific immunoreagents for DNAH17 (HC-β paralogue) and DNAH8 (HC-α paralogue). As suggested by the reviewers, we have now amended the model to indicate a weakening of ODA complex formation (dotted lines/arrows) instead of a complete abrogation as originally indicated (see Figure 4F).

4) Figure 6A and B: Positive controls are required to make sense of negative IP data. For example, if DNAAF2 is confirmed to pull down DNAAF4, but does not pull down ZMYND10, I would be reasonably confident that DNAAF2 does not strongly interact with ZMYND10. Zariwala et al. and Moore et al. identified a biochemical interaction between LRRC6 and ZMYND10. This work fails to recapitulate that interaction. The authors should use the same reagents as Zariwala to resolve this discrepancy. Do they interact or not (or do so extremely transiently)?

We have significantly overhauled Figures 6 and 7, as well as added Figure 4—figure supplement 1 and changed Figure 6—figure supplement 1 to remove ‘negative IP data’, as we concur with the reviewers that these types of studies have limited insight. We refer to the comprehensive DNAAF4 IP-MS dataset generated by Tarkar et al., 2013, which detects DNAAF2 but not ZMYND10 as an interaction partner. More recently, Paff et al., 2017, reported that PIH1D3 interacts with DNAAF2 and DNAAF4 *in vitro*, suggestive of a ternary complex between the three *in vivo*. Their interaction studies also failed to detect ZMYND10 as an interaction partner. Taken together, recent findings suggest that ZMYND10 functions separate to the other DNAAFs. This is why the main focus of this study was the discovery of a novel functional chaperone complex comprising of ZMYND10, FKBP8 and HSP90, the existence of which we have validated in multiple ways. The interaction with LRRC6 is interesting. As shown in our initial submission, we failed to detect ‘endogenous’ interaction between ZMYND10 and LRRC6 by reciprocal IPs during cytoplasmic dynein pre-assembly (now new Figure 6B). Both Zariwala et al., 2013 and Moore et al., 2013 had shown interaction only with overexpression systems. However, LRRC6 is the only ‘DNAAF’ to be significantly down-regulated in our P25 *Zmynd10* mutant testes MS dataset (see Figure 5D), suggesting some type of functional interaction exists. In response to the reviewers' comments, we have now tried a commercially available LRRC6 antibody from Novus (NBP1-82816, as used in the Zariwala paper) but in our hands this antibody fails to recognise endogenous mouse LRRC6 and no enrichment is seen in ZMYND10 pull downs, although a clear interaction with FKBP8 is detected (see Author response image 3).

**Author response image 3. respfig3:** 

We also tested the custom mouse LRRC6 rabbit polyclonal that was a gift from Professor Hiroshi Hamada (used in this paper and panels shown in Author response image 4). Although testes at different developmental stages were used (P24 with synchronised seminiferous tubules and P60 with asynchronised seminiferous tubules), we failed to observe a consistent interaction between ZMYND10 and LRRC6. Instead bands of different molecular weight were enriched in each of the two cases where a potential co-purification was observed (Author response image 4, asterisks). All the while the interaction between FKBP8-HSP90-ZMYND10 is maintained. This suggests that any interaction between these two DNAAFs may be extremely transient in vivo.

**Author response image 4. respfig4:** 

We also requested the human LRRC6-myc plasmid used by Zariwala et al. for our own over-expression studies (new Figure 8A), where we confirm that mouse ZMYND10 does pull-down LRRC6. We also confirm the same patient mutation in mouse ZMYND10 that abrogates interaction with FKBP8, fails to have an effect on the LRRC6 interaction. Furthermore, we show that when LRRC6 is co-expressed with wild type ZMYND10, the amount of FKBP8 precipitated by ZMYND10 is markedly reduced, indicating that ZMYND10 likely forms mutually exclusive complexes with FKBP8 and LRRC6. Again this is consistent with our model of a chaperone relay where several discrete, albeit transient, chaperone complexes oversee folding and stability of discrete dynein subunits, promoting stable interactions between subunits all the while preventing spurious interactions.

**Author response image 5. respfig5:** 

5) The authors provide very interesting data on the role of FKBP8's peptidyl prolyl isomerase activity in dynein assembly and see a decrease in both DHAH5 and DNAI1 stability when this is inhibited. Were these effects specific to these two particular outer arm subunits or is this manifest in others as well e.g. DNAI2 (which associates directly with DNAI1 and for which antibodies are available). In other words, is PPIase activity itself needed for stability of all these components individually, or is their loss the consequence of lack of PPIase activity on one key component of this massive complex?

The reviewer raises a very interesting point, about the specificity of FKBP8 inhibition on stability of a single ‘critical’ component as opposed to a broader effect on several dynein components during cytoplasmic pre-assembly. Our model would have predicted the former, where only stability of HC subunits would be primarily affected. However, we also observed effects on DNAI1 and DNAI2 (Author response image 6). In our 24-hour treatment window, it is possible that DNAI1 and DNAI2 are destabilized secondarily, and as these proteins are much smaller compared to the ~500KDa HCs, we would expect to see faster turnover. Regardless, we show that inhibiting the PPIase activity of FKBP8 phenocopies ZMYND10 loss of function in terms of ODA subunit stability.

**Author response image 6. respfig6:** 

We have also extended this analysis to characterise the effect of DM-CHX on cilia motility as pointed by the reviewers. We used mouse tracheal epithelial cell cultures (mTECs) grown at air-liquid interface and treated at day 3 post-airlift (cytoplasmic pre-assembly starting) with either 100μm DM-CHX or vehicle in differentiation media. Cells were cultured for 14 days (changing every 2 days); whereupon control cultures treated with vehicle showed robust cilial beat. In contrast, no ciliary motility was detected in cultures treated with DM-CHX. We then switched the treatments, such that the control cultures were treated for 7 days with 100μm DM-CHX, which had no effect on motility. Drug treated immotile cultures, when switched to growth medium with vehicle started to display occasional patches of motile cilia. When these cultures were used for wholemount immunofluorescence with DNAH5 and DNAI2 (new Figure 7E), we confirmed FKBP8 inhibition had no effect on fully differentiated motile cilia whereas only newly differentiating cells had high cytoplasmic and some limited axonemal staining in newly elongating cilia, after 7 days release. Dramatically, the majority of cells in the early treatment cohort had no evidence of DNAH5 or DNAI2 in cilia, consistent to lack of ciliary motility. These results strongly support that the specific inhibition of FKBP8 destabilizes dynein subunits preventing their assembly in these cultures and that relieving this inhibition allows partial resumption of the stalled dynein assembly pathway.

6) Some points are overstated. The authors suggest that, "the motile ciliopathy Primary Ciliary Dyskinesia (PCD) should be considered a cell-type specific protein-misfolding disease." Surely, the authors mean that only some forms of PCD should be considered as being caused by protein misfolding.

As suggested by the reviewers, we have amended the wording and tempered our assertion to highlight that PCD subtypes characterised by loss of ODAs and IDAs due to defects in their cytoplasmic preassembly should be considered as cell type specific protein misfolding disorders.